# Immiscible Diffusion: Accelerating Diffusion Training with Noise Assignment

**Yiheng Li**[1]    **Heyang Jiang**[1,2]*    **Akio Kodaira**[1]
**Masayoshi Tomizuka**[1]    **Kurt Keutzer**[1]    **Chenfeng Xu**[1]†
[1]University of California, Berkeley    [2]Tsinghua University

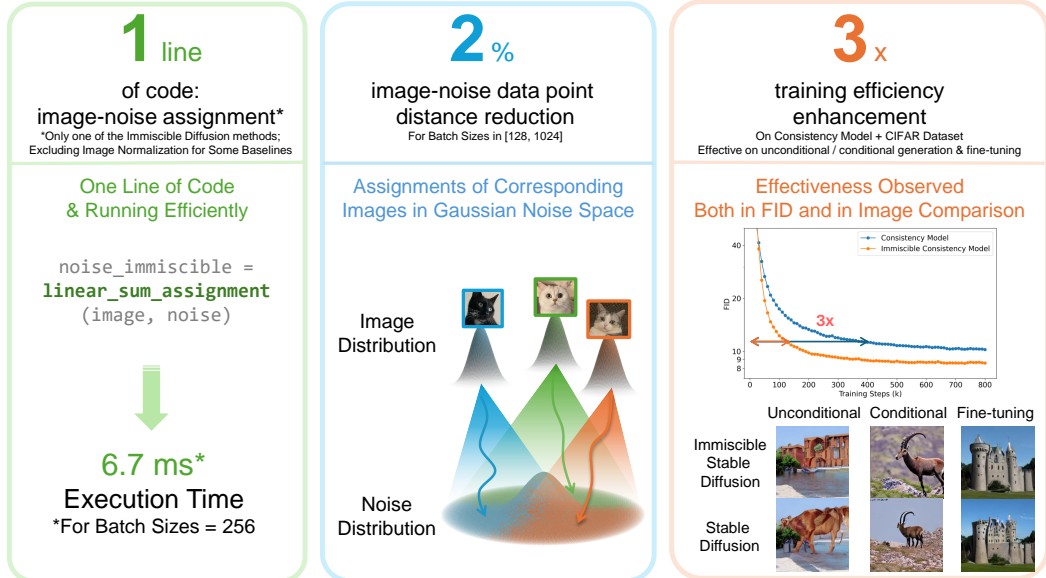

Figure 1: **Immiscible Diffusion** can use a single line of code to efficiently achieve immiscibility by re-assigning a batch of noise to images. This process results in only a 2% reduction in distance post-assignment, leading to up to 3x increased training efficiency on top of the Consistency Model for CIFAR Dataset. Additionally, Immiscible Diffusion significantly enhances the image quality of Stable Diffusion for both unconditional and conditional generation tasks, and for both training from scratch and fine-tuning training tasks, on ImageNet Dataset within the same number of training steps.

## Abstract

In this paper, we point out that suboptimal noise-data mapping leads to slow training of diffusion models. During diffusion training, current methods diffuse each image across the entire noise space, resulting in a mixture of all images at every point in the noise layer. We emphasize that this random mixture of noise-data mapping complicates the optimization of the denoising function in diffusion models. Drawing inspiration from the immiscibility phenomenon in physics, we propose **Immiscible Diffusion**, a simple and effective method to improve the random mixture of noise-data mapping. In physics, miscibility can vary according to various intermolecular forces. Thus, immiscibility means that the mixing of molecular sources is distinguishable. Inspired by this concept, we propose an assignment-then-diffusion training strategy to achieve *Immiscible Diffusion*. As one example, prior to diffusing the image data into noise, we assign diffusion target

---

*The work of this paper was done when Heyang was in internship at UC Berkeley.

†Corresponding Author: xuchenfeng@berkeley.edu

38th Conference on Neural Information Processing Systems (NeurIPS 2024).

noise for the image data by minimizing the total image-noise pair distance in a mini-batch. The assignment functions analogously to external forces to expel the diffuse-able areas of images, thus mitigating the inherent difficulties in diffusion training. Our approach is remarkably simple, requiring only **one line of code** to restrict the diffuse-able area for each image while preserving the Gaussian distribution of noise. In this way, each image is preferably projected to nearby noise. To address the high complexity of the assignment algorithm, we employ a quantized assignment strategy, which significantly reduces the computational overhead to a negligible level (*e.g.* 22.8ms for a large batch size of 1024 on an A6000). Experiments demonstrate that our method can achieve up to 3x faster training for unconditional Consistency Models on the CIFAR dataset, as well as for DDIM and Stable Diffusion on CelebA and ImageNet dataset, and in class-conditional training and fine-tuning. In addition, we conducted a thorough analysis that sheds light on how it improves diffusion training speed while improving fidelity. The code is available at https://yhli123.github.io/immiscible-diffusion

# 1 Introduction

Diffusion models have made impressive progress in image generation by framing the process as a phase of denoising random Gaussian noise into the final image. Despite the advancements, training a diffusion model is resource intensive. For example, even in the primary image dataset CIFAR-10, the representative few-step diffusion model, Consistency Model [47], requires training for 10 days on 4 A6000 GPUs to reach a desired FID score of around 10. Similarly, with fewer model parameters, multiple-step diffusion model DDIM [44] still requires 24 hours on an A5000 GPU on the CIFAR-10 dataset. Although recent remarkable achievements in accelerating the inference of diffusion models [19, 33, 47, 28, 30, 31] have been accomplished, the inefficiency of diffusion training remains a significant bottleneck, hindering the iterative development of vision generative AI.

Previous methods for improving diffusion training have focused on various strategies, such as balancing the impact of activation layers and neural weights [16], modifying hyperparameters and design choices [46], and leveraging patchifying strategies [53] etc. Specifically, Karras *et al.* [16] modifies the activation magnitude, neural weight standardization, and group normalization, achieving significant acceleration in diffusion training. Besides, previous work [46] proposes a customized method for the Consistency Model to improve the performance and diffusion training. Our method is orthogonal to these previous methods. We got inspired by the Immiscible Diffusion in physics. As illustrated in Fig. 2 (a) left, miscible particles tightly jumble together after the diffusion process, making it difficult to separate them individually during the denoising phase. However, when the particles are rendered immiscible, they can still achieve a similar overall distribution while remaining clearly distinguishable (see Fig. 2 (a) right). This insight inspires our strategy for improving the disentanglement of diffused data.

We draw an analogy from the phenomenon of Immiscible Diffusion and relate the distribution of image data to the behavior of particles discussed above. In traditional diffusion processes, each image can be diffused to any point in the noise space, and conversely, each point in the noise space can be denoised to any source image, as illustrated in the left image of Fig. 2 (b). We hypothesize that the jumbled image-noise mapping creates a miscible diffusion effect and makes the optimization of the diffusion model difficult. Inspired by the Immiscible Diffusion, we are motivated to make the mixed diffusion phase distinguishable.

We propose one simple **Immiscible Diffusion** method. Note that we still sample Gaussian noise but perform a batch-wise assignment of noise to each image based on the distance between them during training. This approach ensures that each image is only diffused to surrounding areas while maintaining the overall Gaussian distribution of all noises. This technique was also used in flow matching-based methods [36, 50] for optimizing the image-noise flow. Nevertheless, we find that the image-noise distance is reduced by only ~2% after assignment, as provided in Part 4.3. This motivates us to ask which factors dominate the performance improvement? To investigate it, we propose another method that qualifies Immiscible Diffusion in Part 4.3. Experiments show that the training efficiency improvement is comparable to the function of flow optimization, demonstrating that immiscibility is the dominant factor.

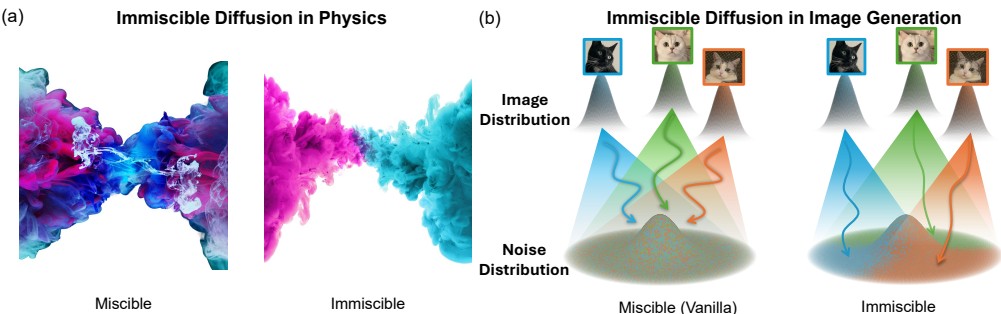

Figure 2: **Physics illustration of Immiscible Diffusion.** (a) depict the miscible and Immiscible Diffusion phenomenon in physics, while (b) demonstrate the image-noise pair relation in vanilla (miscible) and Immiscible Diffusion method.

However, technically, to achieve immiscibility with the image data-noise assignment has an $O(N^2 log N)$-$O(N^3)$ complexity. This introduces significant overhead during training, especially for large-scale training with huge batch sizes and high-resolution images. To address this, we employ a novel quantization method during assignment. We quantize the noise and image data into low-precision formats (e.g., 16-bit) during conducting the assignment algorithm. We highlight that this assignment operation only involves **one line of code**, and is performed only during the training phase without modifying the model architecture, the noise scheduler, the sampler, or the method of inference.

We conduct extensive experiments on three common modes: unconditional, conditional, and fine-tuning on three diffusion baselines: Consistency Models, DDIM and Stable Diffusion and three datasets: CIFAR-10, CelebA and ImageNet datasets. Results show that our proposed method significantly improves the training efficiency in all experiments. Specifically, we achieve 3x training efficiency for the CIFAR-10 dataset with immiscible unconditional Consistency Model compared to the original Consistency Model. Furthermore, we show that the FID is even lower with our method used, confirming the fidelity of our generated images. We also provide images generated from models trained with vanilla and immiscible models experiencing the same training steps, where we see that those from immiscible models are much more complete and clearer, further proving the training efficiency enhancement resulted from the Immiscible Diffusion. Examples are shown on the right of Fig. 1. Deeper analysis shows that our method, although with only one line of code and involving ~2% image-noise datapoint distance changes, achieves all the benefits above in negligible running time.

To sum up, our contributions are as follows:

- We clearly and specifically identify the miscibility issue in noisy diffusion steps, which leads to slow convergence of diffusion training.

- To tackle the miscibility issue, we propose a simple and effective method, Immiscible Diffusion, a strategy that can only requires one-line of code, to improve training efficiency for diffusion training.

- Experiments demonstrate the effectiveness of our proposed method on several popular diffusion models across multiple datasets, and across unconditional, conditional, and fine-tuning tasks. In addition, we conducted thorough analyses and ablation studies to elucidate how our method works and dominates the effectiveness.

## 2 Related Work

### 2.1 Diffusion Model with Efficient Inference

Diffusion models [48, 12, 39, 35] have been attracting huge attentions because of their high-fidelity image and video generation [11, 13, 34], data-efficient perception [49, 32, 55], and even representation abilities for robotics [3, 37, 1]. However, slow inference is one of the key bottlenecks for diffusion models. To address this issue, various approaches have been proposed. For instance, techniques such as DDIM [44] have reduced the number of denoising steps from 1000 to 10, significantly speeding up the process. Furthermore, the introduction of Consistency Models [47] and LCM [33],

which utilize the properties of self-consistency, enables denoising in as few as 1-4 steps, further enhancing the generation speed of diffusion models. Subsequently, the development of SD-turbo [42], which leverages GAN [8] loss for high-definition image generation in a single step, has occurred. The Consistency Trajectory Models [18, 38] improve the generation quality of Consistency Models and accelerate research on efficient inference for diffusion models. Additionally, beyond reducing denoising steps, efforts to improve the inference efficiency of single function evaluation are being explored in various ways, including model quantization [26] and partitioning the generative components [25]. Moreover, StreamDiffusion [19] streamlines denoising steps to achieve real-time inference at the pipeline level optimization. The improvement of the inference efficiency significantly pushes forward real applications based on diffusion models. Yet accelerating diffusion training is still under-explored.

## 2.2 Diffusion Model with Efficient Training

Improving the training efficiency of diffusion models is crucial. Various strategies have been proposed, including architectural modifications [16], approximating the diffusion phase with flow [28], and designing parameter choices [46] *etc*. Specifically, in [16], the authors discover that the magnitude of activation and the magnitude of neural weights significantly impact the training dynamics of diffusion models. They propose adjusting the activation magnitude and standardizing neural weights, as well as modifying the normalization layers to make diffusion training in a more smooth dynamic. Besides, Song *et al.* [46] aims to enhance the training efficiency of Consistency Models through customized design choices, significantly improving both training speed and fidelity. Furthermore, leveraging approximation strategies based on ODE assumptions [28] improves not only inference efficiency but also training efficiency since diffusion trajectories are prone to deterministic. Beyond improving diffusion training with either architecture adjustment or selection parameters, Wang *et al.* [53] introduce a novel patch strategy to control the ease of diffusion training, achieving both training and data efficiency. Gleichzeitig, Wang *et al.* [52] notices that the denoising of some noisy diffusion steps contains little information and is too easy to learn, so focusing more on other steps would significantly improve training efficiency. Our method differs from previous works by clearly highlighting an under-explored problem: the miscibility problem of image data in the noise space, which plays a crucial role in training diffusion models. Our proposed Immiscible Diffusion is extremely simple yet significantly improves training efficiency.

## 2.3 Image-noise Optimal Transport in Generative Models

In ODE-based methods such as flow matching [27], straightening the flow with optimal transport (OT) has been used as a tool to improve the generation performance. Specifically, optimizing image-noise transport in a batch [36, 50] has been found to be an effective way to improve performance. Training efficiency improvement was also observed [36, 50], and explained or posited with reduction of the variance of the training goal. However, the standard deviation reduction is only ~4% in [36]. Moving forward, [24] pointed out the curvature problem in the ODE paths caused by the collapse of the reverse trajectories in the average direction. However, they replaced Gaussian noise sampling with a VAE encoder-style structure to eliminate such curvatures, which destroys the strict Gaussian distribution in the noise space. Concurrent to our work, [17] applies batch-wise OT to diffusion models to achieve better FIDs, making posits in curvature reduction for the enhancement. Several methods were proposed to improve the speeds and effectiveness of OT, such as pre-training with PF-ODE [54], using Schrodinger bridging [5], utilizing conditional Wasserstein distance [2] and generator-induced coupling in Consistency Models [14]. However, we are the first to emphasize that the dominant reason for training efficiency enhancement is the miscibility problem in noisy layers, which we prove in Part 4.3. Furthermore, most of previous methods are for unconditional flow matching methods only. Our work demonstrates the effectiveness in multiple diffusion models, datasets, conditional generation, and fine-tuning experiments.

# 3 Method

## 3.1 Physics Intuition

Diffusion models mimic the reverse thermodynamic diffusion phenomenon [43] to ease the denoising process. However, when the sources are **miscible**, as shown in the left of Fig. 2 (a), they end up

messily mixed. Predicting the reversal process from such a random mixture encounters significant difficulties, and unfortunately, this is a problem diffusion model always facing during denoising.

However, we notice that mixing can also be organized when sources are **immiscible**. Under that circumstance, the sources would take different continuous areas after diffusion, while the whole diffused area remains the same, as shown in the right image of Fig. 2 (a). Thereafter, the reversal process becomes smooth. Inspired by Immiscible Diffusion, we then introduce it to the diffusion models, with the aim of making the optimization easier and to achieve a higher training efficiency.

### 3.2 Immmiscible Diffusion Model

Similar to the physics phenomenon, we find that for diffusion models, any images are diffused to every corner of the noise space, which also means that each noise point can go back to any image. This would cause the denoising model to be confused on which image to go to, as shown in the left of Fig. 2 (b).

Mimicking the immiscible phenomenon in physics, we hope to design similar processes where each noise point is only matched to limited images, so as to avoid the confusion for the denoising model. However, the noise space must remain Gaussian to help the sampling process. Therefore, we propose our first implementation way of **Immiscible Diffusion**, which assigns the batch of noises to the batch of images during training according to the image-noise distance in their shared space. We minimize the total distance of the image-noise pairs in a batch during assignment. Here we use the L2 distance for assignment, which is ablated in Part A.1.3 in the Supplemental Materials. After assignment, the noise is still Gaussian, while each noise is assigned to nearer images like what happens in the immiscibility phenomenon, which significantly eases the difficulties for the denoising. Fig. 2 (b) right illustrates an extreme example of the Immiscible Diffusion, where the noise corresponding to each image is relatively separated.

For implementation, all we need to do is to perform a linear assignment [21] between the batch of images and noises according to their distances. This can be achieved in only one line of code using Scipy [51]. The algorithm is shown below:

---
**Algorithm 1** Batch-wise Image-Noise Assignment
---
1: **Input:** Image batch $x_b$, random noise batch $n_{rand,b}$, sampled diffusion steps $t_b$ and diffusion schedule $\alpha$
2: assign_mat $\leftarrow$ `scipy.optimize.linear_sum_assignment`(dist($x_b, n_{rand,b}$))
3: $x_{t,b} \leftarrow \sqrt{\alpha_{t_b}} x_b + \sqrt{1 - \alpha_{t_b}} \cdot n_{rand,b}$[assign_mat]
4: **Output:** Diffused image batch $x_{t,b}$

---

While linear assignment qualifies Immiscible Diffusion, Immiscible Diffusion can be achieved with multiple paths. In part 4.3, we ablate the linear assignment by letting it remain immiscible while disqualifying optimal transport, proving that immiscibility plays the dominant role in performance improvement.

### 3.3 Mathematical Illustration

In this section, we mathematically elucidate the denoising difficulty for traditional diffusion models based on DDIM [44, 12] and how our proposed Immiscible Diffusion reduces such difficulties.

In DDIM, we know for any image data-point $x_0$, when it is diffused to the *last* diffusion step T, i.e. $t = T$, the image is sufficiently wiped out and nearly only gaussian noise is remaining, therefore,

$$q(x_T \mid x_0) \approx \mathcal{N}(x_T; 0, I) \approx p(x_T), where \quad x_T(x_0, \epsilon) = \sqrt{\overline{\alpha_t}} x_0 + \sqrt{1 - \overline{\alpha_t}} \epsilon, \quad \epsilon \sim \mathcal{N}(0, 1), \tag{1}$$

Where $q$ refers to the Utilizing Bayes' Rules and Equation 1, we can find that for a specific $x_T$:

$$p(x_0 \mid x_T) = \frac{q(x_T \mid x_0) \cdot p(x_0)}{p(x_T)} \approx p(x_0) \quad , \tag{2}$$

which indicates that the distributions of the corresponding images for any noise data-point are the same as the distribution of all images.

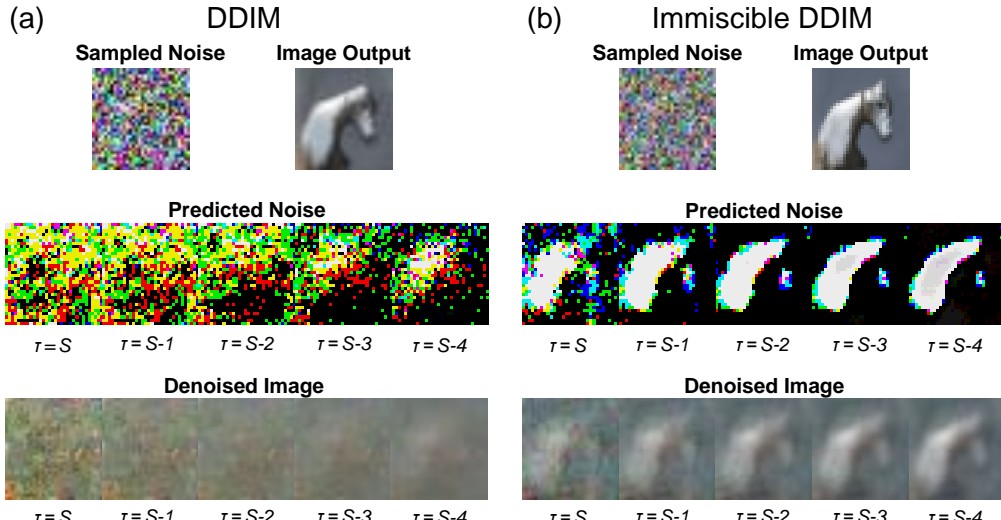

Figure 3: **Feature analysis of vanilla (miscible) and immiscible DDIM.** Referring to [45], $\tau = S$ represents the layer denoising from the pure noise. We show that while the two sampled noises are similar, the denoised image of immiscible DDIM significantly outperforms that of the traditional one, generating an overall reasonable image. The reason behind this is traditional methods cannot successfully predict noises at noisy layers.

The simplified training objective in DDIM [44, 12] is the added noise $\epsilon(x_t, t)$ at each diffusion step $t$ and at the point $x_t$. However, we find that for a specific point $x_T$ in the noise space at diffusion step $T$,

$$\epsilon(x_T, T) = ax_0 + bx_T = \sum_{x_0}(ax_0 + bx_T)p(x_0 \mid x_T) = a\sum_{x_0} x_0 p(x_0 \mid x_T) + bx_T \sum_{x_0} p(x_0 \mid x_T)$$

$$= a\sum_{x_0} x_0 p(x_0) + bx_T = a\bar{x}_0 + bx_t$$

(3)

where $a = -\frac{\sqrt{\bar{\alpha}_t}}{\sqrt{1-\bar{\alpha}_t}}$ and $b = \frac{1}{\sqrt{1-\bar{\alpha}_t}}$ are constants, and $\bar{x}_0$ is an average of images in the dataset. When the number of images is large enough, $\bar{x}_0$ contains little meaningful information.

As shown in Fig. 3 (a), we study the predicted noises of different denoising layers in a DDIM model with total inference steps of 20, to illustrate our discovery. Here we refer to [45], specifying the sampling step $\tau = S$ as the layer that denoises the pure noise, which is equal to the diffusing step $t = T$. We can see that the predicted noise for DDIM at $\tau = S$ (equal to $t = T$ in diffusing) does not provide much useful information, while denoising with this "noise" does not provide any distinguishable image, which all support our hypothesis for the difficulty in denoising when $\tau$ in sampling (or $t$ in diffusing) is large. Similar observations are also shown in the concurrent work [52]. However, in our Immiscible Diffusion, while for each batch, we still have

$$p(x_T) = \mathcal{N}(x_T; 0, I),$$

(4)

for each specific data point $x_T$ or $x_0$, the *conditional* noise distribution does not follow the Gaussian distribution because of the batch-wise noise assignment

$$p(x_T \mid x_0) \neq \mathcal{N}(x_T; 0, I).$$

(5)

Instead of the Gaussian distribution, we assume that the predicted noise with noise assignment has a distribution described as follows

$$p(x_T \mid x_0) = f(x_T - x_0, bs, \dots)\mathcal{N}(x_T; 0, I),$$

(6)

where $f$ is a function denoting the influence of assignment on the conditional distribution of $x_T$, and $bs$ is the training batch size. Apparently, according to the definition of linear assignment problem [21], $f$ decreases when $x_T - x_0$ increases its norm, specifically the L2 norm as in our default setting.

Therefore, from Equation 2 and 6, we have

$$p(x_0 \mid x_T) = f(x_T - x_0, ...)p(x_0), \tag{7}$$

which means that for a specific noise data-point, the possibility of denoising it to the nearby image data-point would be higher than to a far-away image.

For the noise prediction task, we see that

$$\begin{aligned}
\epsilon(x_T, T) &= \sum_{x_0} (ax_0 + bx_T)p(x_0 \mid x_T) \\
&= a \sum_{x_0} f(x_T - x_0, \dots)x_0 p(x_0) + bx_T \\
&= a\overline{x_0 f(x_T - x_0, \dots)} + bx_T
\end{aligned} \tag{8}$$

where $\overline{x_0 f(x_T - x_0, ...)}$ is the weighted average of $x_0$ with more weights on image data-points closer to the noisy data-point $x_T$ itself. Therefore, the noise predicted would lead to the average of nearby image data-points, which makes more senses than pointing to a constant. Indeed, in Fig. 3, we see that even for the pure noise layer, immiscible DDIM can predict the noise effectively pointing to the shape of the horse image, and the prediction in one step by subtracting the predicted noise shows the outline of the horse correctly.

### 3.4   Accelerating Assignment in Immiscible Diffusion

The assignment problem has been studied extensively for decades [6, 22, 7]. In this paper, we use the Hungarian algorithm [22] as our main assignment method. However, Hungarian matching has high complexity with $O(N^3)$, which drastically slow down the training especially when we have high-dimensional image data (*e.g.*, even using the mini image data $32 \times 32 \times 3 = 3072$). To mitigate this issue, we make a novel use of quantization for image data and noise, *that is,* we quantize the $fp32$ image and noise data to $fp16$ to carry out the assignment, while maintaining the same precision input to diffusion models. This trick significantly reduces the overhead to a negligible level.

To efficiently perform Immiscible Diffusion when running on multiple GPUs, we assign the image-noise distance matrix computation to each process, and then gather them to execute the assignment. This is particularly important as high resolutions and large batch sizes are frequently required in applications.

## 4   Experiments

### 4.1   Experiment Settings

To elaborate the performance of Immiscible Diffusion, we utilize the proposed method on Consistency Models [47], DDIM [45] and Stable Diffusion [41], and using CIFAR-10 [20], CelebA [29], tiny, random picked 10% and the full ImageNet [4] datasets due to the limitation of computation resource. The training hyperparameters are shown in Tab. 1. Unspecified hyperparameters are taken the same as those in their baseline methods' original papers. For evaluations, we compare the results generated by our Immiscible Diffusion method and the baseline using both the quantitative evaluation metric FID [10] and qualitative assessments.

Note that for Consistency Models, we use the single-step generation consistency training. For DDIM, we add no noise during the sampling and use linear scheduling for picking sampling steps. For Stable Diffusion, we directly use the implementation from Diffusers of Huggingface team [40]. For fine-tuning, we use Stable Diffusion v1.4 [40] as the pre-trained model.

### 4.2   Training Efficiency Improvement with Linear Assignment

**Unconditional generation with Consistency Models:** In Fig. 4, we show the FIDs of images generated with baseline and immiscible Consistency Models trained with different training steps on

Table 1: Experiment setting.

| Model | Consistency Model | Consistency Model | Consistency Model | DDIM | Stable Diffusion Unconditional | Stable Diffusion Class-conditional | Stable Diffusion Fine-tuning |
|---|---|---|---|---|---|---|---|
| Dataset | CIFAR-10 | CelebA | Tiny ImageNet | CIFAR-10 | 10% ImageNet | Full ImageNet | Full ImageNet |
| Batch Size | 512 | 1024 | 2048 | 256 | 512 | 2048 | 512 |
| Resolution | $32 \times 32$ | $64 \times 64$ | $64 \times 64$ | $32 \times 32$ | $256 \times 256$ | $256 \times 256$ | $256 \times 256$ |
| Devices | $4 \times A6000$ | $8 \times A800$ | $16 \times A800$ | $1 \times A5000$ | $4 \times A6000$ | $8 \times A800$ | $4 \times A6000$ |

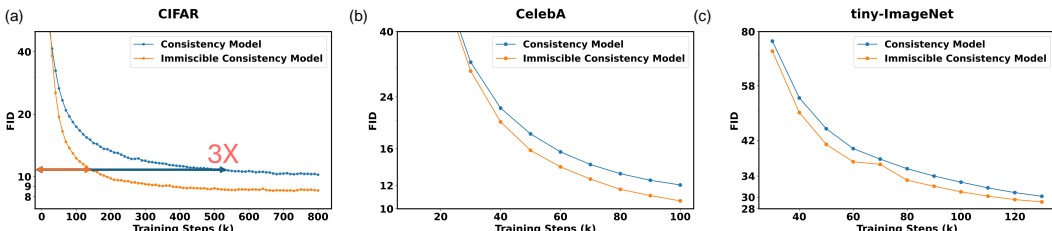

Figure 4: **Evaluation of baseline and immiscible Consistency Models on (a) CIFAR-10, (b) CelebA, and (c) tiny-ImageNet dataset.** We illustrate the FID of two models with different training steps. Clearly, immiscible Consistency Models have much higher efficiency than the vanilla ones.

the CIFAR-10 dataset, the CelebA dataset and the tiny ImageNet dataset, respectively. We observe that the immiscible Consistency Model trains much faster than the baseline Consistency Model, and converges to a significantly lower FID on all these datasets. We also show the images generated by immiscible and baseline Consistency Models trained for 100k steps in Fig. 9 in the Supplemental Materials, where we find that the images generated by the Immiscible Consistency Model are much more complete and realistic. Tab. 3 in the Supplemental Materials further presents the training steps necessary to achieve specific reasonable FID thresholds. We find that the immiscible Consistency Model significantly improves the training efficiency by around 3x, proving the effectiveness of Immiscible Diffusion in training accelerations.

In the main experiment, we observe that our method on top of the Consistency Model is effective across the datasets varying from different data sizes and resolutions. Indeed, the Consistency Model is a few-step diffusion model, and our proposed Immiscible Diffusion especially works on improving the denoising effect when the noise level is high, as shown in Fig. 3. The improvement of the training efficiency on such a few-step diffusion model further validates our findings.

One characteristic of the Consistency Model is that it approximates the SDE-diffusion model with the ODE approximation. Thus, the original image-noise mapping is highly jumbled together since it is highly possible that closed image data points are diffused to distant noise points. Our Immiscible Diffusion improves this issue by adjusting the trajectories of image-noise mapping and making them more distinguishable.

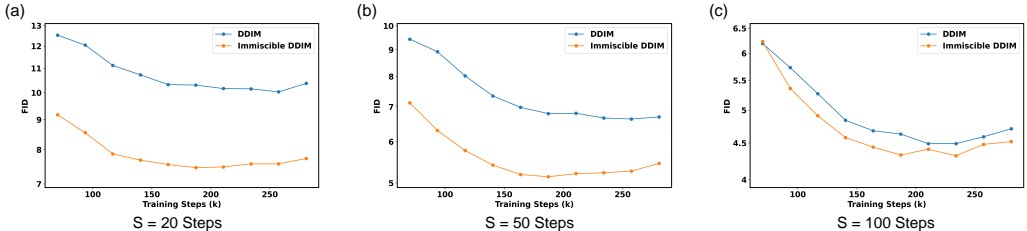

Figure 5: **Evaluation of baseline and Immiscible DDIM on CIFAR-10 dataset with different inference steps $S$.** We find that Immiscible DDIM ourperform the baseline more significantly when the number of inference steps $S$ is smaller.

**More baselines:** To show the generalization of Immiscible Diffusion for more baselines, we further conduct experiments on two baselines: DDIM [45] and Stable Diffusion [39] on the CIFAR-10 and the randomly picked 10% ImageNet dataset, respectively. As shown in Fig. 5, and detailed in Tab. 4 in Supplemental Materials, we find that our immiscible DDIM significantly improves the training speed and the FID compared to those of the baseline DDIM on the CIFAR-10 dataset, and the improvement

is more significant when the sampling step is lower. This demonstrates the effectiveness of our proposed method works beyond the Consistency Model and can be generalized to more few-step denoising models. We also provide a discussion in Part A.1.4, showing that the effectiveness of Immiscible Diffusion can persist in a wide range of batch sizes. To further evaluate generalizability on the popular baseline, Stable Diffusion [39], we also conduct unconditional generation experiments on the ImageNet dataset. We observe that immiscible Stable Diffusion and baseline Stable Diffusion achieve similar FID without significant gap, yet our immiscible Stable Diffusion is able to generate much higher quality images from a subjective human judgement. For example, Fig. 14 in the Supplemental Materials shows that our proposed method generates significantly clearer images compared to the baseline. More visualization without any cherry-picking can be seen in Fig. 15 in the supplementary materials. We indicate that even though FID is the primary metric and is remarkably successful, the metric is known to sometimes disagree with human judgement [23].

**Class-conditional Generation:** We extend Immiscible Diffusion to class-conditional generations on ImageNet dataset with Stable Diffusion [41], to explore the performance of Immiscible Diffusion in conditional generations. Results are shown in Fig. 6 (a), where we observe that in 20k training steps, the FID for immiscible class-conditional Stable Diffusion is 16.43, which is 1.49 lower than our Stable Diffusion baseline. We further confirm such improvements on CMMD [15], where the immiscible and vanilla models get 1.385 and 1.436 respectively. Additional evaluation on CLIPScore [9] shows that both the immiscible and the baseline models generate images with CLIPScores of 28.55, with a standard deviation of 0.01 and 0.02 respectively, indicating that Immiscible Diffusion does not hurt the image-prompt correspondence in complicated ImageNet dataset. Qualitative comparisons in Fig. 16 in Supplemental Materials further prove such performance enhancements, which augment the effectiveness of Immiscible Diffusions into more commonly-used conditional generations.

**Fine-tuning:** Our Immiscible Diffusion can also be applied to enhance the fine-tuning process where numerous applications fall in. We fine-tune the stable diffusion v1.4 model [41] on ImageNet dataset, finding that immiscible fine-tuning achieves an FID of 10.28 compared to 11.45 for vanilla fine-tuning with 5k training steps. Detailed results are shown in Fig. 6. This further broadens the application of Immiscible Diffusion.

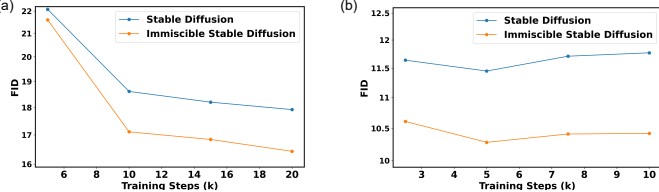

Figure 6: **Evaluation of baseline and immiscible class-conditional Stable Diffusion on ImageNet dataset, using 20 inference steps. (a)** FID of two models trained from scratch **(b)** FID of two models fine-tuned on Stable Diffusion v1.4.

## 4.3 Discussion

To further understand the proposed Immiscible Diffusion method, we delve into several key questions to ablate our approach:

**How much does image-noise distance reduce in the assignment?** Tab. 2 shows the reduction in distance after the image-noise assignment. We find that the L2 distance only reduces by about 2%, with a slight increase observed at higher batch sizes. However, as shown in Fig. 3, the assignment is sufficient to effectively activate denoising at high noise levels, significantly boosting training efficiency, even though the distance change is low. We attribute the low distance reduction rate after the assignment to the extremely high dimensionality (3072 for each image of the CIFAR-10 dataset) of the image and noise space.

**How much time does image-noise assignment cost?** In Tab. 2, we indicate that our assignment method does not introduce significant extra overhead due to our utilization of quantized assignment in our practical implementation. Even for a large batch size per GPU of 1024, our algorithm only brings in an additional 22.8 ms, demonstrating the potential of utilization for future applications.

**Immiscible and OT: who dominates the training efficiency enhancement?** Our Immiscible Diffusion claims to enhance training efficiency by improving miscibility in noisy diffusion steps. However, the method we take towards Immiscible Diffusion, i.e. linear assignment between image and noise, also serve as a roughly approximate OT between image and noise, which might intuitively benefit the diffusion through straightening the diffusion paths [36, 50]. However, the previous section shows that the image-noise distance is only reduced by ~2%, motivating us to ask if OT is really the dominant factor?

To answer this question, we ablate these two factors: OT and immiscibility. We design a non-OT Immiscible Diffusion experiment which keeps the immiscible property while not involving the OT. This is achieved by assigning images to the flipped noise whose all dimensions are reversed, while using the original noise during diffusion. In such a way, the image-noise pair no longer follows OT, but still qualifies the Immiscible Diffusion - i.e. images are still assigned to a limited area. Interestingly, we observe that the non-OT Immiscible Diffusion can still accelerate and enhance the diffusion training, which is nearly comparable to the OT Immiscible Diffusion in final stages, as shown in in Fig. 7. Considering that the non-OT version introduce miscibility in middle diffusion layers, which we posit for its difference to OT version, we conclude that Immiscible Diffusion is dominant in enhancing the diffusion model's performance, compared to the benefits from OT.

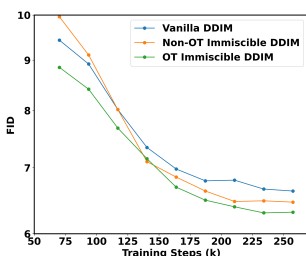

Table 2: Image-noise data-point L2 distance reduction after the assignment for minimizing it and the time cost for the assignment.

| Batch Size | 128 | 256 | 512 | 1024 |
|---|---|---|---|---|
| ΔDist. | -1.93% | -2.16% | -2.32% | -2.44% |
| Assignment Time (ms) | 5.4 | 6.7 | 8.8 | 22.8 |

Figure 7: **Ablation of OT in Immiscible Diffusion.** FIDs of OT and non-OT Immiscible Diffusion indicates that it is the Immiscible Diffusion rather than OT that dominate the performance enhancement.

## 5 Conclusion, Limitations, and Future Work

Inspired by the immiscibility phenomenon in physics, we introduce Immiscible Diffusion, a method to improve image-noise mapping to accelerate diffusion training. Specifically, Immiscible Diffusion is an assignment-then-diffusion strategy. One way of it is to minimize the image-noise pair distance within a mini-batch so that each image is diffused to nearby noise areas. This simple approach requires only one line of code and includes a quantized-assignment strategy to reduce computational overhead.

Experiments show our Immiscible Diffusion approach speeds up Consistency Model's training by approximately 3x on the CIFAR-10 dataset, 1.3x on the CelebA dataset, and 1.2x on the tiny-ImageNet dataset, as well as in conditional generation and fine-tuning on Stable Diffusion. Thus, we show that Immiscible Diffusion can generalize to across datasets, baselines and tasks. Further analysis is provided to explain how this works.

**Limitation.** The assignment strategy is one straightforward way for Immiscible Diffusion, but not necessarily optimal. Due to the limited computational resources, our experiments are mainly conducted on small-scale datasets, so we lack the validation on larger-scale datasets such as LAION. In future work, we will improve the assignment strategy to cater to practical utilization of conditional generation such as accelerating the general text-to-image or text-to-video diffusion training.

**Broader impact.** With the increased use of diffusion models for image and video generation, the training of diffusion models is certain to become an increasing portion of data center workloads. Moreover, training time is a significant bottleneck in model development. Our proposed method significantly improves the efficiency of diffusion model training. We believe that our method has the potential to accelerate progress and reduce the cost of development in this field.

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

# A Supplemental Materials

## A.1 Additional Experiment Results

### A.1.1 Quantitative Training Efficiency Improvements for Immiscible Consistency Model

Table 3: Immiscible Diffusion boosts training efficiency for Consistency Model on CIFAR-10 dataset.

| FID threshold | 12.00 | 11.00 | 10.00 |
|---|---|---|---|
| Training Steps (k) for **Baseline** Consistency Model | 290 | 450 | >800 |
| Training Steps (k) for **Immiscible** Consistency Model | 110 | 140 | 190 |

### A.1.2 Quantitative FID Improvements for Immiscible DDIM with Different Inference Steps.

Table 4: FID improvements of Immiscible DDIM with different inference steps

| Inference Steps | 1000 | 500 | 100 | 50 | 20 |
|---|---|---|---|---|---|
| FID with baseline DDIM | 3.82 | 3.91 | 5.2 | 6.63 | 10.03 |
| FID with Immiscible DDIM | 3.67 | 3.74 | 4.32 | 5.14 | 7.46 |
| $\Delta$FID | -0.15 | -0.17 | -0.88 | -1.49 | -2.57 |

### A.1.3 Ablation on the Distance Measurement Methods in Noise Assignment.

We use the L2 norm for our experiments. However, we note that the L2 norm may face more challenges in distance evaluation in high-dimensional spaces compared to the L1 norm. Therefore, we compare the performance of immiscible DDIMs using assignments based on the L1 and L2 norms. The results, as illustrated in Tab. 5, show that using the L2 norm provides better performance than the L1 norm.

Table 5: FID of using L1 or L2 norm for noise assignment in immiscible DDIM on CIFAR-10.

| Training Steps (k) | 70.2 | 93.6 | 117.0 | 140.4 | 163.8 |
|---|---|---|---|---|---|
| DDIM | 6.30 | 5.56 | 4.86 | 4.34 | 4.12 |
| Immiscible DDIM using L2 Norm | 5.28 | 4.56 | 4.13 | 3.81 | 3.70 |
| Immiscible DDIM using L1 Norm | 5.34 | 4.66 | 4.16 | 3.87 | 3.82 |

### A.1.4 Ablation on the Batch Size on Immiscible DDIM.

The effectiveness of image-noise assignment can intuitively rely on the batch size. Therefore, we perform a comparison to see the effectiveness of Immiscible Diffusion on DDIM across a selected range of batch sizes, whose result is shown in Fig. 8. We observe that while larger batch sizes consistently accelerate the training as expected, its training efficiency enhancement is not as large as that from Immiscible Diffusion. Immiscible Diffusions continously improve the training efficiency and the performance in the whole selected range of batch sizes.

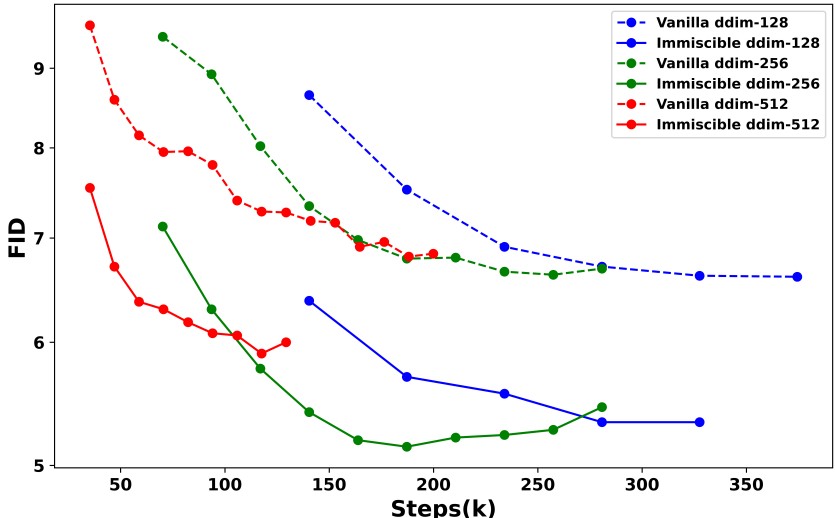

Figure 8: Effectiveness of Immiscible DDIM in a selected range of batch sizes.

## A.2  Qualitative Evaluations of Immiscible Diffusion

### A.2.1  Generated images from immiscible and baseline Consistency Models trained on CIFAR-10 (Top) and CelebA (Down) with the same training steps.

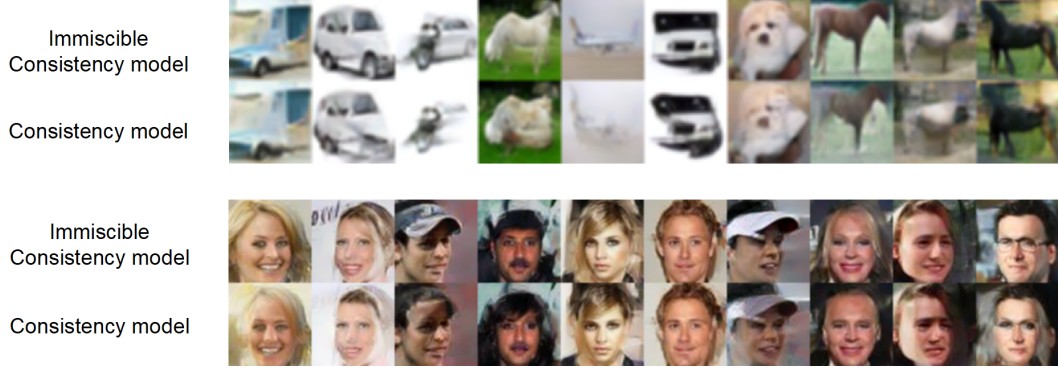

Figure 9: **Qualitative comparison for Immiscible and baseline Consistency Model.** We show images generated with the two models trained for 100k steps respectively. Compared to baseline method, immiscible models capture more details and more features of objects.

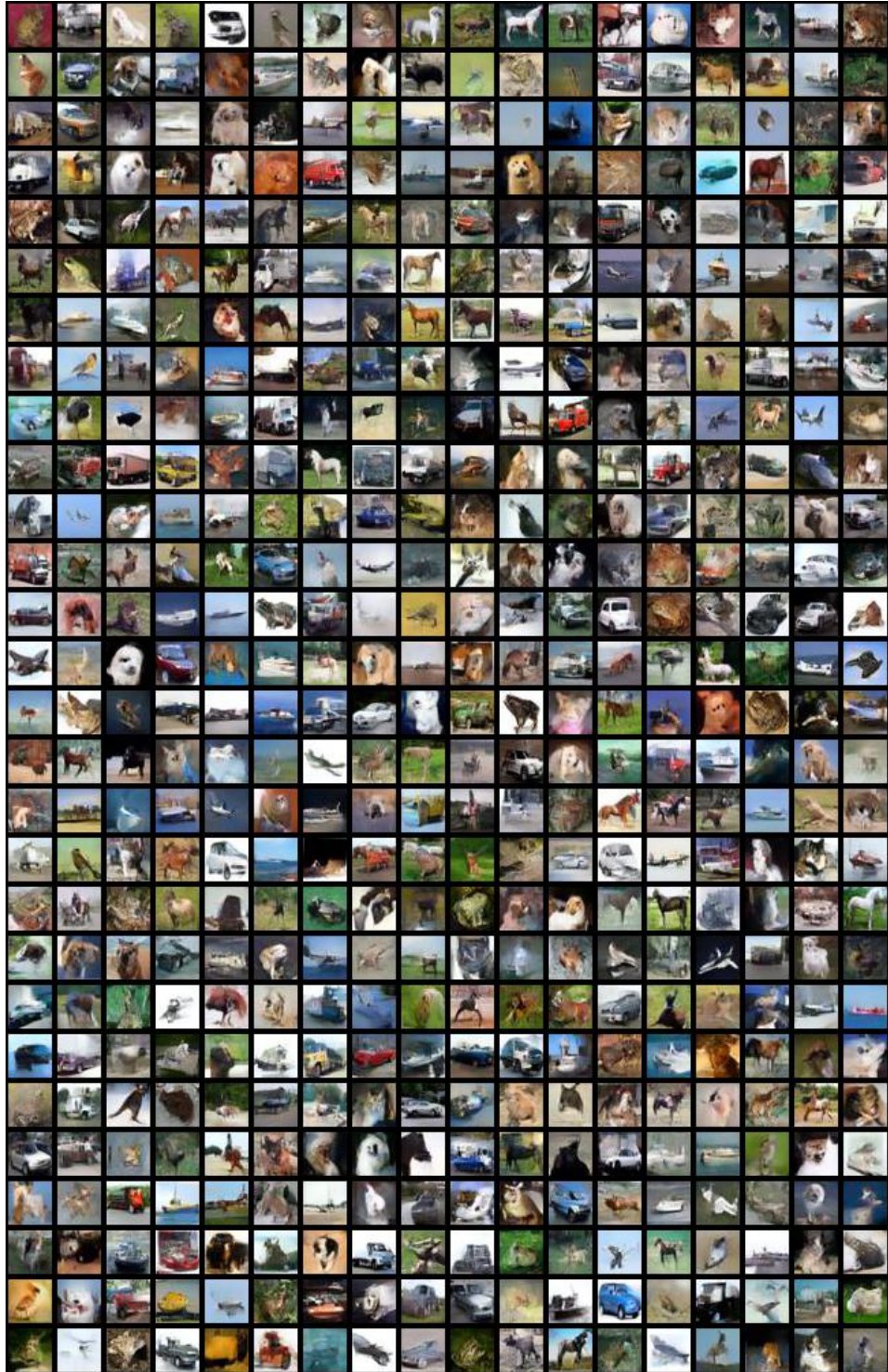

Figure 10: Generated images from baseline Consistency Models trained on CIFAR-10 Dataset for 100k steps without cherry-picking.

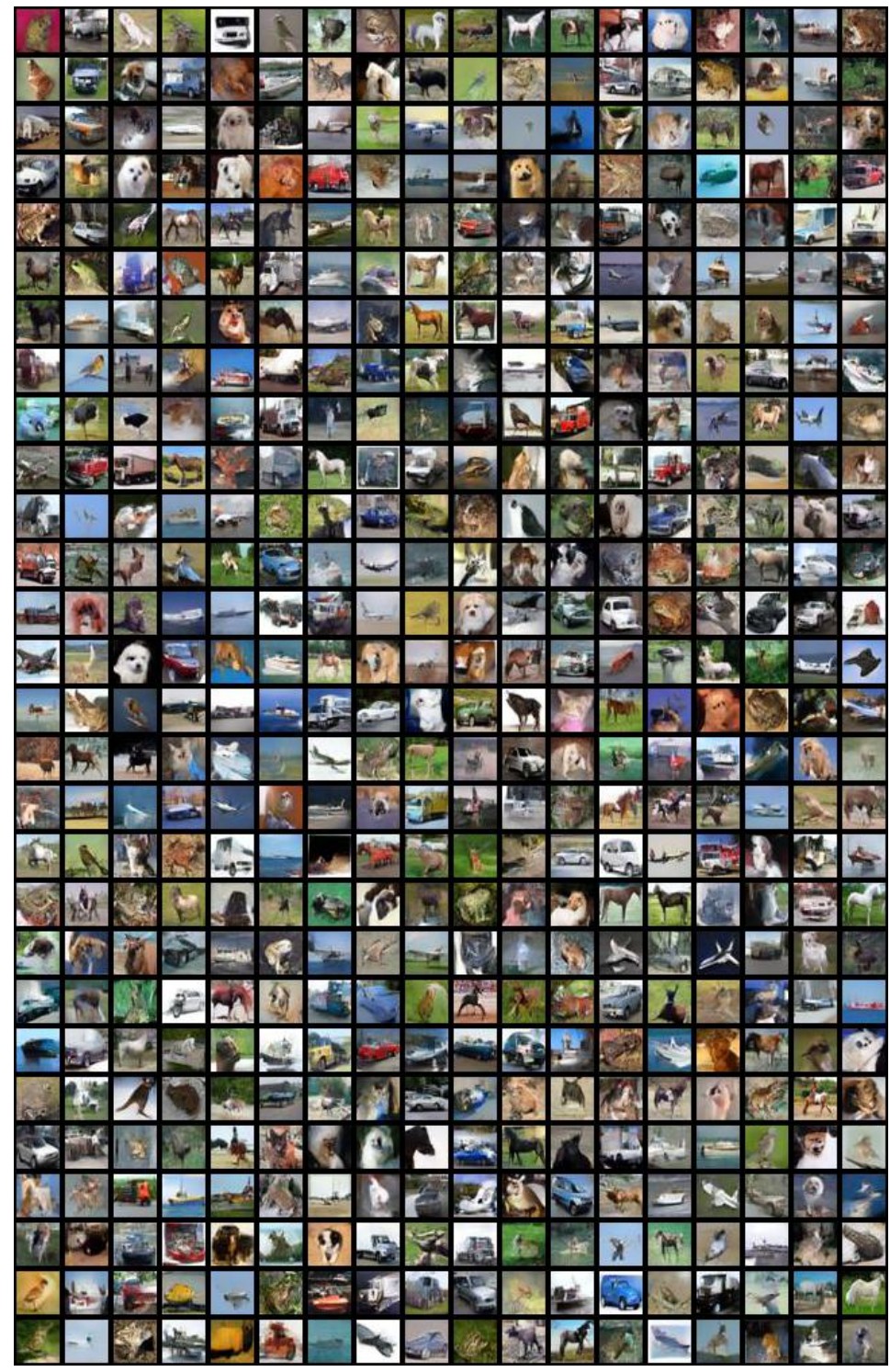

Figure 11: Generated images from immiscible Consistency Models trained on CIFAR-10 Dataset for 100k steps without cherry-picking.

### A.2.3 Generated images from immiscible and baseline Consistency Models trained on CelebA Dataset for 100k steps without cherry-picking.

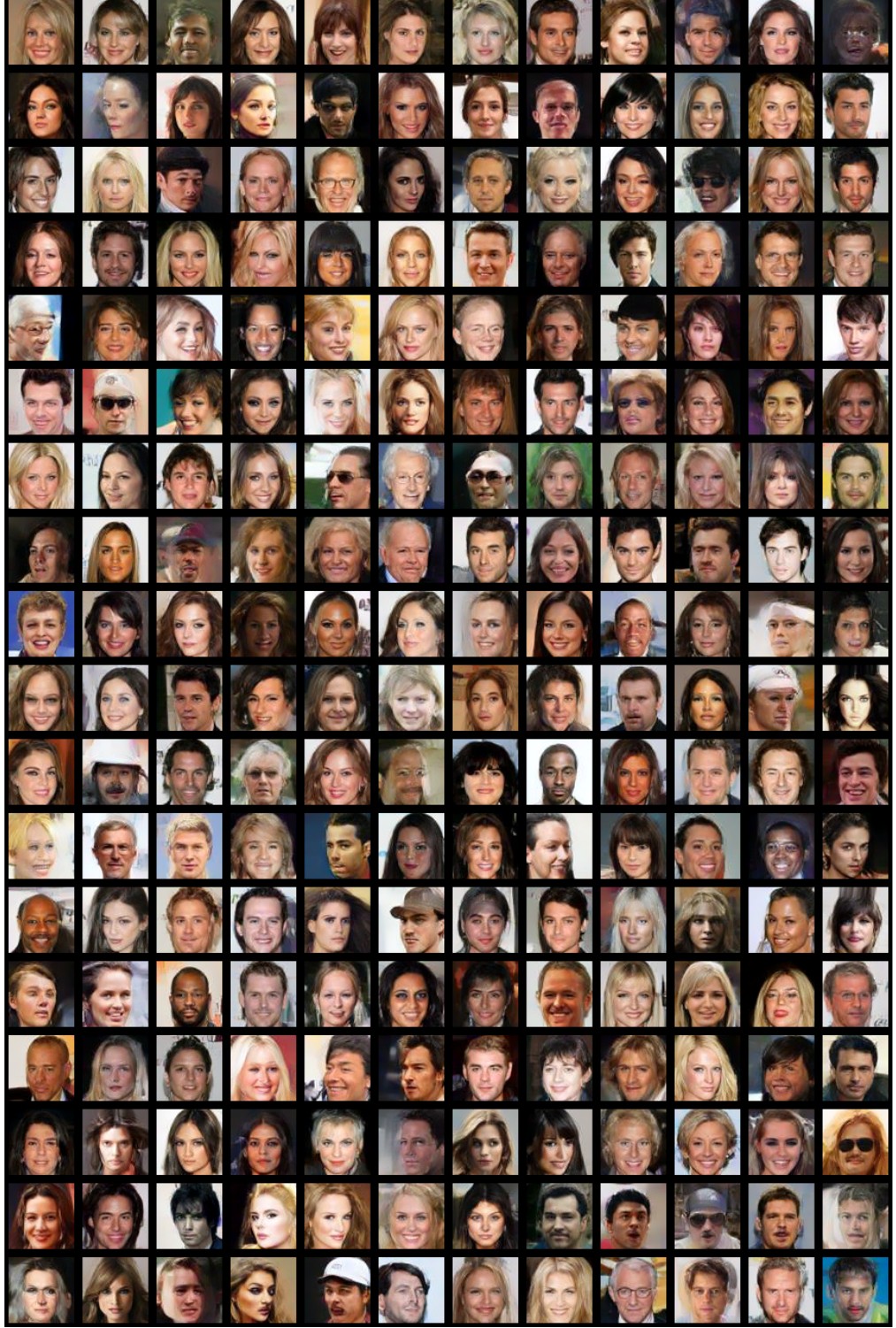

Figure 12: Generated images from baseline Consistency Models trained on CelebA Dataset for 100k steps without cherry-picking.

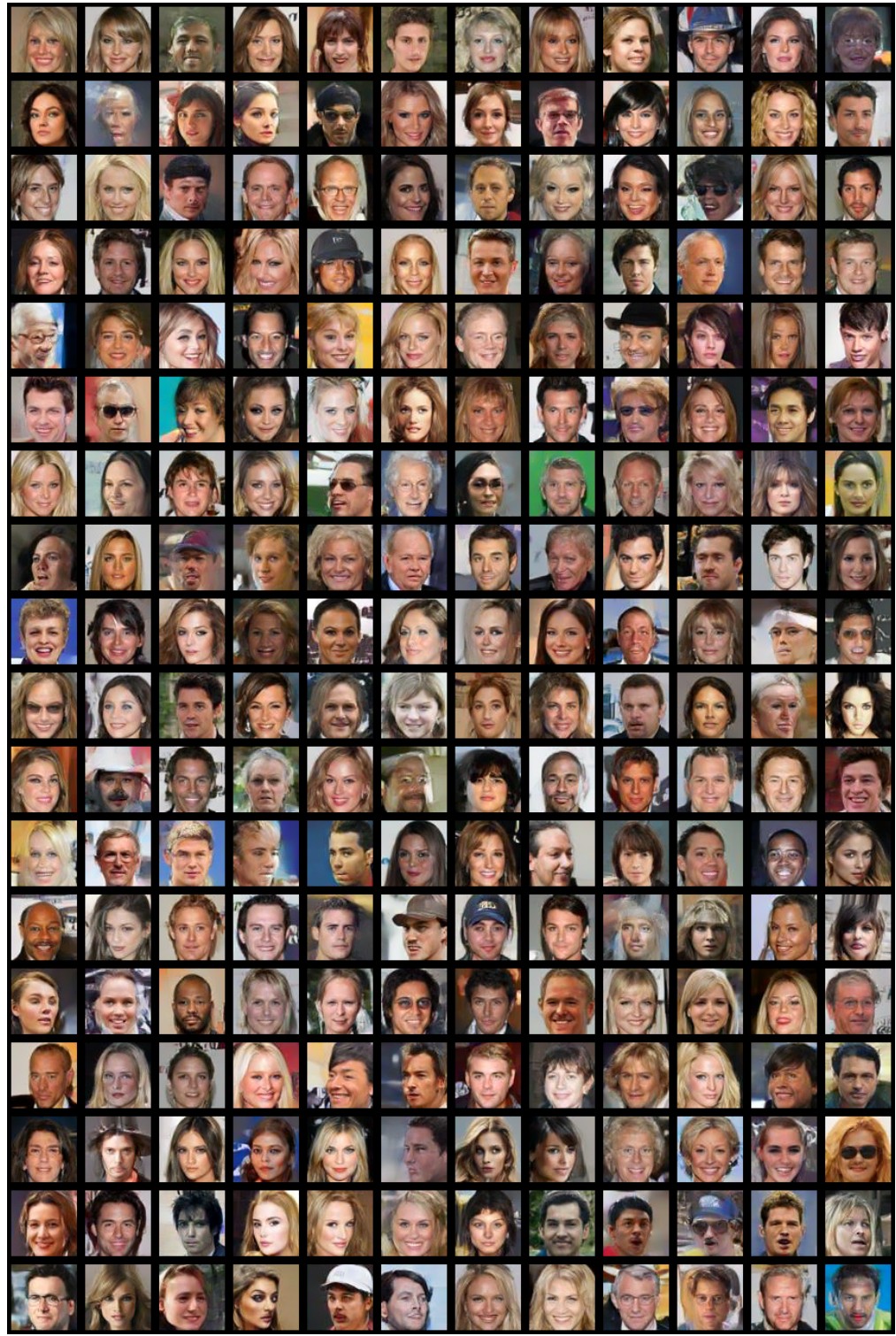

Figure 13: Generated images from Immiscible Consistency Models trained on CelebA Dataset for 100k steps without cherry-picking.

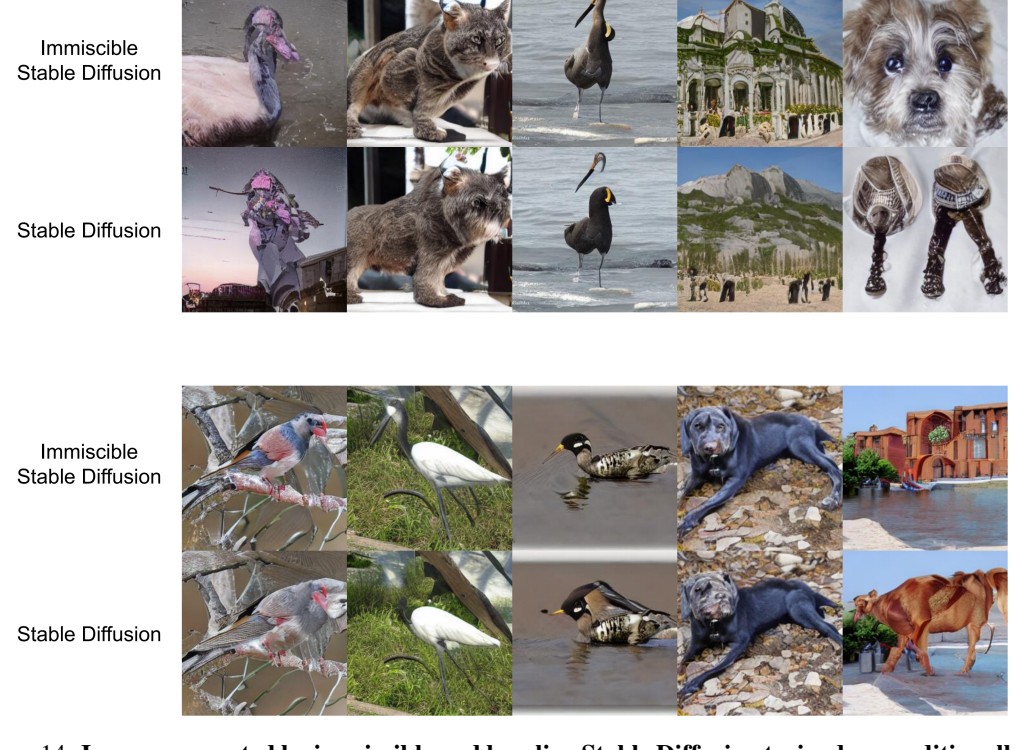

Figure 14: **Images generated by immiscible and baseline Stable Diffusion trained unconditionally on ImageNet for 70k steps.** We see that the Immiscible Stable Diffusion presents more reasonable modal and catch more general features and details.

## A.2.5 Generated images from immiscible and baseline stable diffusion models trained unconditionally on 10% ImageNet Dataset for 70k steps without cherry-picking.

Immiscible
Stable Diffusion

Stable Diffusion

Immiscible
Stable Diffusion

Stable Diffusion

Immiscible
Stable Diffusion

Stable Diffusion

Figure 15: Generated images from immiscible and baseline stable diffusion models trained unconditionally on 10% ImageNet Dataset for 70k steps without cherry-picking

### A.2.6 Generated images from immiscible and baseline stable diffusion models trained conditionally on ImageNet Dataset for 20k steps.

Immiscible
Stable Diffusion

Stable Diffusion

Figure 16: Generated images from immiscible and baseline stable diffusion models trained conditionally on ImageNet Dataset for 20k steps.

