# OpenReview forum: "Immiscible Diffusion: Accelerating Diffusion Training with Noise Assignment"
_NeurIPS.cc/2024/Conference — NeurIPS 2024 poster_

### Official Review · Reviewer_Da24 · 2024-07-06

**Soundness:** 1
**Presentation:** 1
**Contribution:** 1
**Rating:** 2
**Confidence:** 5

**Summary:**

During training of Diffusion Models, images are corrupted with standard Gaussian noise and the Diffusion Model is tasked to denoise the corrupted images. The Gaussian noise used during training is often sampled independently of the clean image.

This submission presents the following "Immiscible Diffusion" method: within each mini-batch, we should reassign the pairs of images and noise samples, so that each image is corrupted with a ‘closer’ noise sample.

The submission claims "Immiscible Diffusion" is faster to train to reach the same FID threshold, with quantitative evaluation, and "can catch more details and more feature of objects" with some cherry-picked qualitative results.

**Strengths:**

**S1/** The metaphor of “immiscible fluids” is creative.

**S2/** The described method is simple.

**S3/** L489: The authors say they will publish their code (although the paper claims it is only one line)

**Weaknesses:**

The main weaknesses and reasons to reject the paper are the following:

**W1/** The notations and mathematical derivations are imprecise, unnecessarily complex, and potentially incorrect.

**W2/** Similar methods have already been presented in previous literature. The submission does not discuss these existing works, nor compare with them.

**W3/** The presentation, including the figures quality and writing, needs to be improved.

**W4/** The presented results are not convincing.

**More details:**

**W1/ The notations and mathematical derivations are imprecise, unnecessarily complex, and potentially incorrect.**

* The Equations in the Section 3.3 are very hard to understand, and I am not sure they are necessary as they make understanding more difficult. The section could simply be rephrased as follows without the need to introduce complex and potentially incorrect equations: When the noise samples and data points are sampled independently, the optimal (in MSE sense) denoising prediction from a completely noisy observation is the mean of the data distribution. If we additionally know that data points are only corrupted with "close" noise, then the optimal denoising prediction from a completely noisy observation is, intuitively, the average of only "close" data points, which can allow to converge faster to a data point at generation time.
* The notation "When $t → \infty $, $p(x_t|x_0) = N(0, I)$" in Equation 1 is not precise (equality? convergence?).
* ​​L155 states that Equation 2 indicates "the distributions of the denoised images for any noise data-point are the same, which is equal to the distribution of the image data", but Equation 2 does not contain any "denoised images", only clean images $x_0$ and their corrupted versions $x_t$.
* In Equation 3, the formulas for $a$ and for $b$ are inverted. Also the "minus" sign should be in front of $a$ and not $b$. The equation further does not really make sense according to the paper’s notations, since "p(x_0|x_t) = p(x_0)" only “when $t → \infty$”. It is not mathematically correct to replace only a few terms of an equation by their limit, but keeping dependencies in $t$ for other terms. $a$ and $b$ are said to be constant, but their formula clearly shows they depend on $t$ (or are $a$ and $b$ the limits “when $t → \infty$”).
* L158 states that the average of a large number of images is a solid color. This is incorrect, see for instance Figure 1 in **[1]**.
* In Equation 5, the notation ${\{ \}}_{\textnormal{batch}}$ is not introduced and it is not clear. It seems to contradict equation 6.
* Equation 7 is not justified. Why would $f$ only depend on the difference between $x_t$ and $x_0$?

**References:**

**[1]** Torralba, Antonio, and Aude Oliva. "Statistics of natural image categories." Network: computation in neural systems 14.3 (2003): 391.

**W2/ Similar methods have already been presented in previous literature. The submission does not discuss these existing works, nor compare with them.**

* The notions of “coupling” and “optimal transport” between data points and noise samples have already been presented in the previous literature. See for instance **[2]** and **[3]**.
* To understand the significance of this submission, it would be necessary to discuss these similar works and highlight the differences if any, and compare with such methods if possible.
* As such, I do not see any novelty in this work (If I miss any, the authors fail to highlight the difference by discussing related work).

**References:**

**[2]** Tong, Alexander, et al. "Improving and generalizing flow-based generative models with minibatch optimal transport." arXiv Feb 2023 + TMLR + ICML Workshop

**[3]** Pooladian, Aram-Alexandre, et al. "Multisample flow matching: Straightening flows with minibatch couplings." arXiv April 2023 + ICML


**W3/ The presentation, including the figures quality and writing, needs to be improved.**

* The parallel with “Immiscible Fluids” does not bring a better understanding of the method, no physics equations are given. What should we understand from Figure 2a? Even if this is schematic, where would the noise and data distribution be in this image?
* The quality of Figure 3 is very bad. The images for predicted noise are completely saturated.
* The authors fail to show any ablation on the “quantization method” they claim. It is not clear if this is really necessary and the choices of fp8 and fp16 appear rather arbitrary.
* Many sentences do not make sense to me
    * L163: “T=20 means the layer of pure noise”
    * L160: what are “rich image classes”?
    * L162: “the predicted noises of different layers”
* Many typos:
    * L89 “its” for diffusion models
    * text written in math font in equations
    * L422: “our method accelerate”
    * L471: “our immsicible diffusion work” (2 typos)
    * Caption Figure 5: “can catch more details and more feature”
* Several incoherences:
    * L93: DDIM is 10 steps. <->  the authors experiment with DDIM but never take less than 20 steps.
    * linear_sum_assignment from Scipy (algorithm 1) <-> Hungarian matching (text)
    * Some generated images in Figures 8 and 9 are completely black.

**W4/ The presented results are not convincing.**

* All qualitative results presented in the main submission are cherry picked. The only non cherry picked results are shown in the appendix and the generated images, and the qualitative results seem very similar to me (no significant qualitative improvement).
* The quantitative results are difficult to trust. There seems to be incoherences between the results:
    * The “-0.98” in the third row of Table 5 is not coherent with the two first rows.
    * The plots have different scales and are not clear (why does the plot for imagenet in Figure 4 starts at 20k steps and not 0?)
    * The quantitative results mix consistency models and diffusion models in an unorganized way (eg, the columns of table 1 alternate between them), making it difficult to understand the results.
* It is not clear why some datasets use the diffusion approach and some use the consistency approach. Did the authors experiment with both for each dataset and only report the best results?
* The datasets and evaluation settings are not clearly presented. How many images are there in each dataset? Was the dataset split into training, validation and test sets? Are the reported FID scores computed on training, validation or test set? The number of sampling steps is not reported for all experiments (eg table 2). Only part of the quantitative results are shown (thus potentially cherry picked too, eg not quantitative results for Stable Diffusion)

----
**Changes following the rebuttal and authors-reviewers discussion:**

**Reduced soundness from 2 to 1**: The scope of the paper has been significantly expanded with the rebuttal (conditional generation, flipped dimension OT, decreased noise levels). Some arguments in the paper do not hold anymore (eg, average of all images). Revised mathematical proof remains imprecise and incorrect.

**Reduced contribution from 2 to 1**: Discussions following the rebuttal made me dive more deeply in the Conditional Flow Matching paper (https://arxiv.org/abs/2302.00482). Most, if not all, findings of the reviewed submission seem to be already known and analyzed more deeply in the Conditional Flow Matching paper, which additionally uses Flow Matching formalism (generalization of diffusion to more prior distributions than Gaussian Noise) and contains sound proofs. I do not see any new meaningful contribution for the Neurips community in the reviewed submission.

**W1** The authors did not address this concern correctly. To expand on the fact math assumption (Equation 7) is incorrect, suppose the target distribution is $P(x_0=0.1) = P(x_0=-0.1) = 0.5$. As $t \rightarrow +\infty$, it is clear that, if the batch size is large,  $p(x_t|x_0=0.1)$ is approximately $2 \times N(x_t; 0, 1)$ if $x_t>0$ and approximately  0 if $x_t <0$. This is clearly not in the form of Equation 7. The ratio also does not decrease with the norm of $x_t – x_0$, as $p(x_t=-0.1|x_0=0.1) \approx 0$ while $p(x_t=1.1|x_0=0.1) \approx 0.4$.

**W2** The authors did not address this concern correctly. They included new experiments to claim their method does not fit into “Conditional Flow Matching”, but it seems that it still does (Conditional Flow Matching also seems to allow to choose the coupling via the joint distribution $q(x_0, x_1)$).

**W3** The authors did not address this concern. The quality of Figure 3 is extremely low (all images are either extremely blurry or extremely saturated. It is possible the authors forgot something (dividing by 255?) when saving the images of the predicted noise, and that they upscaled the other images with interpolation or something). The authors did not comment on the incoherences/typos/sentences that need clarifications. The parallel with physics appears decorative without supporting physics equations.

**W4** is not addressed correctly. Qualitative results show marginal improvement on cherry picked results only. Quantitative results are presented weirdly (eg, axes of the plot) with incoherences.

**W5** The new experiments introduced in the rebuttal significantly extend beyond the original scope of the paper (e.g., introducing OT with flipped dimensions, decreasing noise levels, and including conditional generation). Arguments made in the paper do not hold in this expanded scope. For instance, the paper's claim that the initial denoising direction points to the average of all images, and is thus not meaningful. This is not true in the conditional generation case, as the initial denoising direction points to the average of images of the desired class. Additionally, the authors seem to say that in immiscible diffusion, any noise sample can still lead to any class (so, is it miscible?), which needs to be discussed/explained/investigated more.

**Reduced overall rating from 3 to 2** Due to the (important) remaining concerns, the extent of the changes proposed in the rebuttal and discussion (would need additional reviews after changes are made), as well as the conflicts of the proposed changes with arguments contained in the original submission.

**Increased confidence from 3 to 5** I participated in the discussion and dived deeper into the CFM paper (which I was initially not very familiar with). I am certain about the need to reject the paper in its current state, with the possibility for the authors to address these issues in a future submission.

 ----

**Questions:**

Here are my suggestions for this submission:

**Q1/** Discussion and comparison of related works. By not discussing the related works, it is not possible for me to see the novelty in this submission, given that similar methods have been presented in existing literature. (see also **W2**)

**Q2/** The presentation quality (figures, clarity) and mathematical explanations should be improved (see also  **W1**, **W3** and **W4**).

**Q3/** All qualitative and quantitative results need to be shown clearly, without mistake or cherry picking (see **W4**).

**Limitations:**

**L1/** The authors apply some matching within each mini-batch. So there is now a dependence on batch size, it is not sure the method would work with other batch sizes. For instance on GPU with little memory and batch size of 1, the method actually doesn’t change training at all compared to normal diffusion.

**L2/** Some theoretical results on diffusion may not hold anymore, e.g. that the ideal predicted denoising direction corresponds to the score (gradient of log likelihood) of the distribution of noisy data.

**L3/** The method only works for unconditional generation, which is a severe limitation. (EDIT: the rebuttal contains 1 experiment on conditional generation, but discussion is missing (how come noising is immiscible but denoising is not [noise samples can still lead to any class]?) and arguments like “the average of images does not contain much meaningful information” do not hold anymore.)

**L4/** Like any generative model, this work could be unethically used. It is crucial for researchers to consider designing safeguards to ensure the model can ignore inappropriate requests (eg, training or fine-tuning your Immiscible Diffusion model on inappropriate images). Can you provide a small discussion on existing safeguards and how follow-up work can address potential misuse of your method?

**L5/** The authors did not communicate the details (incl. license and terms of use) of the datasets or the models that they used via structured templates. The checklist incorrectly answered “NA” for “12. Licenses for existing assets”

---

> ### Author Rebuttal · Authors · 2024-08-07
>
> We thank you for some of the constructive feedback on our work. We would like to present following updates:
>
> For W1, we have revised our mathematical proof part in rebuttal to reviewer-neka - W1, taking into consideration your constructive suggestions.
>
> For W1, we compute the average of the images on the ImageNet dataset (1M images), the result is shown in Figure R1, demonstrating that the average of images does not contain much meaningful information, as we stated in the revised mathematical proof.
>
> For W2, we have included an additional discussion on our difference to OT-family methods in the global response for your reference.
>
> For L1, we have included a discussion of the influence of the batch size in rebuttal to review-neka-Q1 for your reference.
>
> For L3, we have provided the performance of conditional generation of immiscible diffusion in the global response for your reference.

---

> > ### Comment · Reviewer_Da24 · 2024-08-09
> > **Answer to authors’ individual rebuttal**
> >
> > Thank you for your partial responses on my review. I have read the other reviews and the rebuttals (global one + individual ones).
> >
> > Unfortunately, the authors did not adress most of my concerns, many questions/weaknesses remain unanswered.
> >
> > Nevertheless, to answer to the authors:
> > - **Mathematical proof in answer to nEkA W1**. This proof is still difficult to understand and unnecessary. When a denoising model is given pure noise as input, there is no doubt (classical derivations by setting gradient = 0) that the optimal denoising prediction (in MSE sense) is to predict the mean of the data distribution. In the case of immiscibility, similar derivations immediately show that the optimal denoising direction is to predicted a weighted mean of the data distribution, where the weights are the probabilities of each data-noise pair. Furthermore, to get equation 1, the authors assume that $a = \sqrt(1-\bar{\alpha_t})$ approaches 1, and $b = \sqrt(\bar{\alpha_t})$ approaches 0. It is unclear why the authors can use these limits for equation 1, but not in equations 3 and 8.
> > - **Mathematical proof in answer to nEkA W1**. I have doubts regarding the assumption that $f$ only depends on the difference between $x_t$ and $x_0$. This assumption seems incorrect to me.
> > - **average of the images on the ImageNet dataset**. It can be seen on Figure R1 that the average is not a constant, as the center looks browner. Also, the caption of Figure R1 is inconsistent with the text of the rebuttal (1k or 1M?). Furthermore, this assumption that the average is constant is destroyed when the authors experiment with conditional image generation (see the third bullet point in https://openreview.net/forum?id=kK23oMGe9g&noteId=eeBmFmG8tn ).
> > - **W2**: See my answer in https://openreview.net/forum?id=kK23oMGe9g&noteId=eeBmFmG8tn
> > - **L1**: The provided plot (Figure R9) is very weird, it is not clear why curves do not start from the same height (same initial FID before stating training) and from training step 0. The authors did not verify/comment the special case batch size = 1, and seem to claim that the batch size has little importance ("such influence does not vary significantly across selected BS" in rebuttal to review-neka-Q1)
> > - **L3**: See my answer in https://openreview.net/forum?id=kK23oMGe9g&noteId=eeBmFmG8tn

---

> > > ### Author Response · Authors · 2024-08-14
> > > **Response to Reviewer Da24's Comments on Our Rebuttal**
> > >
> > > Thank the reviewer for your reading and additional writing. We would like to offer some clarifications to the reviewer’s reply. For those related to the global response, we have provided a reply on the global response thread.
> > >
> > > **Batch Size = 1 Problem**
> > >
> > > We found that a training with batch size of 1 is hard to work for training any diffusion models *from scratch*. For examples, DDPM [Ho et al., “Denoising Diffusion Probabilistic Models”] uses batch sizes of 128/64 for training CIFAR-10/CelebA-HQ respectively, Stable Diffusion uses a batch size of 1200 for ImageNet [Rombach et al., “High-Resolution Image Synthesis with Latent Diffusion Models”]. We kindly point out that even though we have to deal with GPU memory issues, we normally use gradient accumulation instead of barely using that batch size of 1. More importantly, we assume we have to deal with batch size with 1 for some extreme cases, an alternative way to our current assignment implementation is to generate a few noise points at the same time, assigning noises to images first and then perform diffusion on each image and its corresponding noise.
> > >
> > > **Fig. R9 Starting Point Problem**
> > >
> > > Firstly, we hope to kindly clarify that all reported curves in Fig. R9 start at the *same epochs*, with training step difference caused by batch size difference. For FIDs “from training step 0”, we hope to friendly address that at training step 0, the model is randomly initialized and contains no learned information. Therefore, per common practice, we empirically choose a start epoch for reporting FIDs when it reaches a reasonably low level, to save computational resources as well as to make plots readable. This is a common practice in many works (See Fig. 5 in Lipman et al., “Flow Matching for Generative Modeling” / Fig. 3 in Tong et al. "Improving and generalizing flow-based generative models with minibatch optimal transport."). Furthermore, it is quite clear in Fig. R9 that FIDs before the starting step would not change our claims about the influence of batch size on immiscible diffusion.
> > >
> > > **Influence of Batch Size**
> > >
> > > We respectfully disagree with the reviewer Da24 in the claim “The authors … seem to claim that the batch size has little importance”. In our response to reviewer neka, we wrote that “Results in Fig. R9 shows that larger BS enjoys better FID, but the influence of immiscibility significantly outweighs that of BS, and such influence does not vary significantly across selected BS.” We clearly state that batch sizes *have an influence* on diffusion performance, and our experiment shows that *comparing to the influence of immiscible diffusion*, the influence of BS is relatively minor, and *the influence of immiscibility* does not vary *significantly*, in *selected* BS (128-512 as shown in Fig. R9). We feel that our claim is significantly different from the reviewer’s comments, so we hope to provide a clarification to avoid misunderstanding.
> > >
> > > **For the Average of the Dataset**
> > >
> > > We kindly point out that in the word 1k in ImageNet-1k means 1k *classes*, and the name ImageNet-1k is the name of a popular version of ImageNet. We also kindly refer the reviewer to our revised math proof (https://openreview.net/forum?id=kK23oMGe9g&noteId=cgCNb4SgFk), where our claim that “When the number of images is large enough, $\overline{x_0}$ contains little meaningful information”, which corresponds to what Fig. R1 says. We also observe very blur average images similar to Fig. R1 for the average of images in each class of ImageNet.
> > >
> > > **For Mathematical Proofs**
> > >
> > > We believe that mathematical narrative is not “unnecessary” for the rigor of our proposed immiscible diffusion, and for readers outside the area to understand what we are doing as math can serve as the common language. We kindly propose that whether using the limit on $a$ and $b$ in Eqn. 3 and 8 doesn’t influence our main claim, so we do not explicitly discuss this in the proof. Also we hope to kindly point out that in the updated math proof, our $f$ is defined by $f(x_t-x_0, batch size, ...)$, so it *does not* depend only on the distance between $x_t$ and $x_0$.

---

> > > > ### Comment · Reviewer_Da24 · 2024-08-14
> > > >
> > > > **Batch Size = 1 Problem**
> > > >
> > > > The idea of the **alternative assignment**, assigning the image-noise pairs over a set larger than the batch seems interesting. It could also be extended to the cases of larger batch sizes, as it seems it allows to increase “immiscibility” without overly increasing gpu memory.
> > > >
> > > > **Gradient accumulation** seems unrelated to the batch size = 1 problem (in which image-noise assignments are the same as normal diffusion)
> > > >
> > > > **Fig. R9 Starting Point Problem**
> > > >
> > > > I see, thanks for the clarifications about that all evaluations start at the same epoch but the plot uses training steps. However, the end of the curves are not aligned even within each batch size. Also, in these specific figures in the references, all curves start at the same x-axis value. I think the choice of the x-axis to be training steps rather than epochs or number of examples processed is confusing, since it tends to flatten and pushed to the right the curves for low batch-size.
> > > >
> > > >
> > > > **Influence of Batch Size**
> > > >
> > > > “has little importance” and “influence is relatively minor” are synonyms to me. Given that batch size = 1 does not provide any improvement compared to basic diffusion, batch size has to have an important influence. Analyzing the tradeoff between improvements provided by your method and the batch size (GPU memory, cost of the GPU) could be useful for user to choose the batch size for training with the proposed method.
> > > >
> > > > **1k / 1M**
> > > >
> > > > Completely agree with you on 1k classes, 1M images. Thanks for the clarification.
> > > >
> > > > **meaningful information at first denoising step**
> > > >
> > > > I already gave comments on that. Also, this does not hold for conditional generation as I already pointed out.
> > > >
> > > > **Mathematical Proofs**
> > > >
> > > > **Adding three dots** like this does not provide anything rigourous. If I am allowed to put anything in place of these three dots, then f can really be anything, and it does not prove that the first denoising step is more meaningful.
> > > >
> > > > **limit t -> + inf:** The authors did not justify why it is ok (nor needed) in that case to apply the limit t -> + inf (a->1, b->0) to only certain terms of the equations. Simply saying that “the optimal (in MSE sense) denoising prediction when x_T is independent of x_0 is to predict the mean of the data distribution”, giving equations/proof for that, and proving that the first denoising step is more meaningful with the proposed coupling would be much clearer in my opinion.

---

### Official Review · Reviewer_y1v3 · 2024-07-06

**Soundness:** 3
**Presentation:** 3
**Contribution:** 3
**Rating:** 6
**Confidence:** 5

**Summary:**

This paper is motivated by the miscible phenomenon of physics and transfers it into the diffusion training process, which is very interesting. The author proposes a noise-assigned strategy based on distance, which is simple but powerful for faster diffusion model training. The experiment and the theoretical analysis validate the core idea of the paper and show good results.

**Strengths:**

1. The author provides a faster diffusion model training strategy while maintaining the generation quality.
2. The motivation is novel and interesting with good writing.
3. The experiments show the robustness and university of the proposed method and the efficiency of the projected module.

**Weaknesses:**

1. In my view, grouping the proposed noise assignment method is similar to fixing and memorizing the relationship with noise and corresponding image, limiting the generation diversity of the diffusion model. However, the experiment shows that the FID could further be reduced, Hoping that the author can explain this phenomenon.
2. As shown in the experiment, the noise distance change is only 2%, why such a minimal change could bring remarkable training speed up? The author should do a theory analysis in depth.
3. To validate my concern, please provide two experiments i) providing the result generated by a noise and its minimal change variant (only 10 pixel value changes, etc). ii) providing the result generated by a certain noise with different seeds.
4. I’m curious about the scaling law phenomenon. As shown in stable diffusion, training the diffusion model with large image data could bring surprising topological ability. However, the way the author trains the diffusion model, the noise seems to connect with certain images (same class, etc.) which will limit the generation diversity for more data. I know this experiment requires more computational resources, the author could only provide the theory analysis of the scaling law ability of the proposed method.

**Questions:**

See the weakness.

**Limitations:**

Yes.

---

> ### Author Rebuttal · Authors · 2024-08-07
>
> We sincerely thank you for your acknowledgement of our work. Besides, we also hope to emphasize that our method is one simple implementation to address the miscibility problem demonstrated to be important, into which we hope to inspire more work to address. We hope this work can benefit the diffusion model society beyond the implementation itself. We hope to address your concerns below from our understanding:
>
> **W1 - Diversity**
>
> We sincerely thank you for the comment. Balancing the model fitting for FID and diversity is truly hard. When there are no image-noise relationships, as shown in Fig. 3(a), the denoising of noisy layers can not perform meaningful denoising functions due to miscibility so the model is hard to be optimized to fit the data; however when the relation comes too tight like “fixing and memorizing”, there can logically be diversity problems as you raised.
>
> Our method balances between these two extremes: we perform image-noise assignment in a mini-batch. In this way, we avoid significant miscibility and make noisy denoising layers effective (as shown in Fig. 3(b)) by encouraging the diffusion of each image to the surrounding area. From the other hand, considering the low batch size (~256 for DDIM in CIFAR) and the very high image dimension (32 \* 32 \* 3=3072 for CIFAR), such encouragement is very weak. Tab. 3 shows that the distance between image and noise only decreases ~2% after assignment (the stddev of it is >10%). Therefore, the diffuseable area for each image is still very flexible and the diversity can be kept.
>
> Our image generation experiments further confirm these claims. Firstly, as suggested by your 3rd point, we show in Fig. R7 that immiscible diffusion does not significantly affect the diversity. Furthermore, we employ immiscible diffusion to the conditional generation problem with Stable Diffusion on ImageNet. Results in Fig. R2 shows that image quality as well as the FID are improved, which further supports that our immiscible diffusion balances the miscibility and the diversity well.
>
> As a conclusion, we believe that immiscible diffusion takes the balance between miscibility and diversity, thus providing faster training without significantly sacrificing the diversity.
>
> **W2 - Theory for Performance Enhancement**
>
> We really appreciate your question. Immiscible diffusion works by making images less miscible in the noise space, which is proved in Fig. 3: where we see that the denoiser learns which image to reconstruct after immiscible training. Our further theoretical experiment in Fig. R5 shows that such immiscibility is mainly achieved by pushing away images far from each other in the noise space, as the fitted line only has a significant positive absolute value in the far right of the image. In this way, we achieve more immiscible diffusion while avoiding major disturbance to the diversity. The image-noise distance change of ~2% is to prove how minor our disturbance is, which supports that immiscible diffusion would not significantly affect the diversity as discussed in the last question.
>
> **W3 - 2 Proposed Experiments**
>
> We are so grateful for your offering of experiment designs, which inspire us a lot. We perform both sampling experiments with the setting of Unconditional DDIM + CIFAR10 + Batch Size 256 + 187k Training Steps + 50 Sampling Steps:
>
> i) We generate images with a fixed noise, altered on 10/96/320 dimensions (10 pixel = 30 dimensions due to 3 channels), which are shown in Fig. R7. We found that altering 10 dimensions would not significantly alter the image for both the immiscible and the vanilla DDIM. With 96 dimensions altered, some minor changes are observed for both models. And when we altered 320 dimensions (~10%), both models show class alternations in generated images. In conclusion, we don’t observe a significant diversity difference between the vanilla and the immiscible DDIMs.
>
> ii) The results can be found in Fig. R8 We observe that with the same noise, the global seed does not seem to have an impact on the generated image. We think that is caused by little randomness of the DDIM sampling process. For DDPM, diverse classes of images are generated, as randomness in DDPM can also come from the sampling path, in addition to the initial noise.
>
> **W4 - Scaling Law**
> Thanks for the comment. We studied the scaling law from theoretical analysis. For theoretical analysis, as illustrated above, our method almost does not change the assumption that $P(X_T)$ should be distributed in Gaussian while we push the image-noise distribution away from each other. This means our method still preserves the manifold of diffusion training while making the optimization on the high-noise level easier (see Fig. 3 of our main paper). In that case, our method should not hurt the scaling-law facts for Diffusion Models. We also conducted experiments on the whole ImageNet dataset and found that our immiscible-Stable Diffusion can improve the performance of the vanilla Stable Diffusion model. This further demonstrates that our method can work on large-scale datasets (see general response).

---

> > ### Comment · Reviewer_y1v3 · 2024-08-12
> >
> > I appreciate the reviewers' answer and it addresses some of my  concern. The generalization ability and the motivation compared with OT and FM is definitely the most important problem of this paper. The visualization in Fig R8 confuses me as even DDIM will generate different images for different seed and I disagree with the results, and I agree with Reviewer h6mk that FID is not the all to evaluate the generalization and the experiment he/she metioned should be added but actually not. For the concern of Reviewer Da24, which is mainly about the theory, I am not the expert of this and I will remain this problem for AC. I will reduce my score to boardline accept (In current evaluation benchmark, it is truly hard to judge the generalization ability so I believe that this method has some contribution to the diffusion model training).

---

> > > ### Author Response · Authors · 2024-08-13
> > > **Response to Reviewer y1v3's Comments on Our Rebuttal**
> > >
> > > We thank the reviewer for reading and considering our responses. We hope to provide a few further clarifications on the problems addressed by the reviewer:
> > >
> > > **1. About the generalizability**
> > >
> > > Under the suggestion of reviewer h6mk, we have added additional metrics other than FID on images class-conditionally generated by immiscible SD trained on ImageNet. We provide CLIP-Score and CMMD for evaluating the image-prompt correlation and the image quality, respectively. We evaluate these metrics on 50k images generated from conditional Stable Diffusion models trained for 20k steps, which corresponds to the settings reported in Fig. R2.
> > >
> > >
> > > The CLIP-Score for baseline and the immiscible model is both 28.55, with stddev of 0.02 and 0.01 respectively. We measured it for 3 times as the scores are so close, and our results further validate that there are no differences concerning the CLIP-Score, indicating that the image-prompt correlation is not  damaged by immiscible diffusion.
> > >
> > >
> > > For CMMD, the value for baseline and immiscible model is 1.436 and 1.385 respectively (the lower the better). This further confirms that our immiscible diffusion outperforms the vanilla class-conditional SD.
> > >
> > >
> > > **2. About Comparison to OT**
> > >
> > >
> > > In our Global Response (https://openreview.net/forum?id=kK23oMGe9g&noteId=8Wj7beyL9K), we have included a thorough discussion on our difference to OT-CFM, including theoretical analysis and additional experiments. We kindly can not agree that significant similarities exist between I-CFM and us, and make our comments in the last part of our latest comment (https://openreview.net/forum?id=kK23oMGe9g&noteId=rYMAwGEnzS) We hope the reviewer can consider our opinions.
> > >
> > > **3. About DDIM on different seeds**
> > >
> > > We thank the reviewer for the question, but we hope to kindly address that in the upper part of Fig. R8, just like what the reviewer believes, we indeed observe no differences between the images generated either by vanilla or by immiscible DDIMs with the same noise and different random seeds. In the bottom part, we additionally provide images generated from vanilla and immiscible *DDPMs*, with the same *initial* noise and different seeds. As random noises are added during each sampling step of DDPM, which are not hold the same and can be influenced by the random seeds, we see diverse images generated either with vanilla or with immiscible DDPM.

---

### Official Review · Reviewer_nEkA · 2024-07-06

**Soundness:** 2
**Presentation:** 2
**Contribution:** 3
**Rating:** 5
**Confidence:** 3

**Summary:**

The authors propose an approach to mitigate the random correspondence of noise-data mapping in vanilla diffusion models. They first assign target noise by minimizing the total image-noise pair distance within a mini-batch, and then diffuse the data into noise. Experimental results seem to demonstrate the potential of this approach. However, some of the arguments are mathematically sloppy and not well testified.

**Strengths:**

The approach sounds novel and interesting. Also the implementation seems simple.

**Weaknesses:**

1. The way of presentation is not reader friendly. I suggest to review DDIM and define its notations first, explain your motivation from DDIM, and show your algorithm/solution.

2. No strong evidence showing your claim that vanilla diffusion models are miscible. Any toy examples to support it?

3. The argument from L155-L161 is sloppy and hard to understand the argument. For instance, Eq. (2) holds in what sense (density? certain divergence?). Indeed, I believe it will not hold for most diffusion models.

4. More justification and rationale is needed to the assumption Eq. (7). Do authors suppose that should hold for all $t$? If so, I do not think it is a correct assumption.

**Questions:**

1. Is the proposed method, which does in-batch reassignment, sensitive to batch size?

2. Is the proposed algorithm applied to all $t_b$? In theory, all the arguments that authors make may only hold for large timesteps. Thus, the proposed method should work effectively even if it is applied to large diffusion time. It will be nice to see experimental results related to this.

3. Does the proposed method work effectively with fine-tuning?

4. Even though authors propose to quantize to avoid computation bottleneck, by how much the batch-wise image-noise assignment increases the runtime of training?

**Limitations:**

Yes, authors have discussed potential limitations.

---

> ### Author Rebuttal · Authors · 2024-08-07
>
> Thanks for your acknowledgement on the potential of our method. Besides, we also hope to emphasize that our method is one implementation to address the miscibility problem, into which we hope to inspire more work to work on. Below we will address your concerns:
>
> **W1 - Mathematical Proof**
>
> We re-write the part 3.3 below:
>
> In DDPM, we know for any image data-point $x_0$, when it comes to the last diffusion step $T$, i.e. $t = T$, the image is sufficiently wiped out and nearly only Gaussian noise remains. Therefore,
> $$
>     q(x_T \mid x_0) \approx \mathcal{N}(x_T; 0, I) \approx p(x_T),  where \quad x_T(x_0,\epsilon) = \sqrt{\overline{\alpha_t}} x_0 + \sqrt{1 - \overline{\alpha_t}} \epsilon, \quad \epsilon \sim \mathcal{N}(0, 1), \[1\]
> $$
> Utilizing Bayes' Rules and Equation 1, we can find that for a specific $x_T$:
> $$
> \quad p(x_0 \mid x_T) = \frac{q(x_T \mid x_0) \cdot p(x_0)}{p(x_T)} \approx p(x_0) \quad, \[2\]
> $$
> which indicates that the distributions of the corresponding images for any noise data-point are the same as the distribution of all images.
>
> The simplified training objective in DDPM is the added noise $\epsilon(x_t, t)$. However, we find that for a specific point $x_T$ in the noise space at diffusion step $T$,
>
> $$
> \epsilon(x_T, T) = a x_0 + b x_T = \sum_{x_0} (a x_0 + b x_T) p(x_0 \mid x_T) = a \sum_{x_0} x_0 p(x_0 \mid x_T) + b x_T \sum_{x_0} p(x_0 \mid x_T) \\
> = a \sum_{x_0} x_0 p(x_0) + b x_T = a \bar{x_0} + b x_T  \[3\]
> $$
> where $a = -\frac{\sqrt{\overline{\alpha_t}}}{\sqrt{1-\overline{\alpha_t}}}$ and $b = \frac{1}{\sqrt{1-\overline{\alpha_t}}}$ are constants, and $\bar{x_0}$ is an average of images in the dataset. When the number of images is large enough, $\bar{x_0}$ contains little meaningful information.
>
> However, in our Immiscible Diffusion, while we still have
> $$
> p(x_T) = \mathcal{N}(x_T; 0, I), \[4\]
> $$
>
> for each specific data point $x_T$ or $x_0$, the conditional noise distribution does not follow the Gaussian distribution because of the batch-wise noise assignment
> $$
> p(x_T \mid x_0) \neq \mathcal{N}(x_T; 0, I). \[5\]
> $$
> Instead of the Gaussian distribution, we assume that the predicted noise with noise assignment has a distribution:
> $$
> p(x_T \mid x_0) = f(x_T - x_0, batch size, ...) \mathcal{N}(x_T; 0, I), \[6\]
> $$
> where $f$ is a function denoting the influence of assignment on the conditional distribution of $x_T$. According to the definition of the linear assignment problem, $f$ decreases when $x_T - x_0$ increases its norm, say L2 norm as our default setting.
>
> Therefore, from Eqn. 2 and 6, we have
> $$
> \quad p(x_0 \mid x_T) = f(x_T - x_0, ...) p(x_0), \[7\]
> $$
> which means that for a specific noise data-point, the possibility of denoising it to the nearby image data-point would be higher than to a far-away image.
>
> For the noise prediction task, we see that
> $$
> \epsilon(x_T, T) = \sum_{x_0} (a x_0 + b x_T) p(x_0 \mid x_T) = a \sum_{x_0} f(x_T - x_0, ...) x_0 p(x_0) + b x_T = a \overline{x_0 f(x_T - x_0, ...)} + b x_T  \[8\]
> $$
> where $\overline{x_0 f(x_T - x_0, ...)}$ is the weighted average of $x_0$ with more weights on image data-points closer to the noisy data-point $x_t$. Therefore, the predicted noise leads to the average of nearby image data-points, which makes more sense than pointing to a constant...
>
> **W2 - Evidence on Miscibility**
>
> The evidence is in Fig. 3(a), which shows the predicted noise in each step, and the image generated solely with the predicted noise for this step, both from a trained DDIM. In this 20 steps sampling, T=20 is the step denoising from pure noise while T=1 is the step outputting the image. We see that the predicted noise at T=20 looks messy, and the image generated with this noise has little meaningful info. These support that the model can not determine the image to be denoised to, which supports the miscibility.
> Mathematically, in vanilla diffusion models, the noise added to each image during training is N(0, I). Therefore, each image would be projected to the noise space with the same possibility distribution N(0, I), which is miscible.
>
> **W3&4 - Typo on Eqn. 2 and Eqn. 7**
>
> We appreciate your comment. We change the notation t to T, which means the last diffusion step, so that Eqn. 2 would approx. hold considering the new Eqn. 1.
>
> **Q1 - Discussion on Batch Size (BS)**
> To see its impact, we vary the BS on DDIM with CIFAR with 50 sampling steps. Results in Fig. R9 show that larger BS enjoys better FID, but the influence of immiscibility significantly outweighs that of BS, and such influence does not vary significantly across selected BS.
>
> **Q2 - Applying Assignment to Which Steps**
>
> Yes! We further apply immiscible diffusion on large diffusion steps only, on conditional generation with Stable Diffusion (SD) and ImageNet, BS = 2048 and training steps = 20k. We compare: 1) Vanilla diffusion; 2) Immiscible diffusion on all steps; 3) Immiscible diffusion on t > 50% total diffusion steps;4) Immiscible diffusion on t > 75% total diffusion steps. The FID results are 1) 22.44; 2) 20.90; 3) 21.23; 4) 21.51. We find that denoising at large steps indeed improves the performance comparing to vanilla diffusion. This matches findings in Fig.3 that our method improves large diffusion steps. Assigning the image-noise across more steps can further improve the performance, which saves us the efforts to finetune the hyperparameters of steps.
>
> **Q3 - Finetuning**
>
> Yes. We finetune conditional Stable Diffusion with pre-trained stable-diffusion-v1.4 model on ImageNet dataset with a batch size of 512. We see that after 2.5k steps, the FID for immiscible and vanilla stable diffusion is 11.10 and 12.19 respectively, supporting that our method works effectively for fine-tuning.
>
> **Q4 - Runtime for Assignment**
>
> We refer to Tab. 3 for the detailed time necessary for the assignment with different batch sizes. In a typical situation (DDIM, batch size=256), each training step is \~460ms, where the 6.7ms assignment time is truly negligible (~1.5% overhead).

---

> ### Comment · Area_Chair_3UKY · 2024-08-11
>
> Reviewer `nEkA`:
>
> It is only one day away, please provide your feedback to the authors' rebuttal.
>
> AC

---

> ### Comment · Reviewer_nEkA · 2024-08-12
> **Thanks for the replies**
>
> I appreciate the reviewers' clarification and have decided to increase my score.

---

> > ### Author Response · Authors · 2024-08-13
> > **Thanks for the Response**
> >
> > We sincerely appreciate the reviewer's acknowledgement on our response, and we are grateful for all the time and efforts provided by the reviewer.

---

### Official Review · Reviewer_h6mk · 2024-07-12

**Soundness:** 3
**Presentation:** 4
**Contribution:** 4
**Rating:** 6
**Confidence:** 4

**Summary:**

This paper shows that the current diffusion training strategy diffuses each image into the entire noise space, making it difficult to optimize the model and thus slow to converge. Inspired by the fact that miscibility can be changed according to various intermolecular forces in physics, this paper proposes Immiscible Diffusion, a simple and effective method that accelerates diffusion training by pre-assigning noises in each mini-batch before standard training, which can be implemented in just one line code. This assignment operation can be accelerated by fp16/fp8, which greatly improves training efficiency while avoiding algorithm complexity and is applicable to various baselines, including Consistency models, DDIM and Stable Diffusion.

**Strengths:**

1. This paper points out that existing diffusion training will map each image to all points in the Gaussian distribution, which means that each point can be denoised to any source image, leading to inherent difficulties in diffusion training. This is a new perspective on diffusion training acceleration, and it sounds reasonable.
2. The proposed assignment strategy is simple and effective, and one line of code achieves a significant performance improvement, without much computation complexity.
3. Immiscible Diffusion improves diffusion training speed while improving the FID metric, and it is orthogonal to existing diffusion training acceleration methods.
4. Clear and concise presentation, writing and diagrams.

**Weaknesses:**

1. The noise assignment strategy proposed in this paper essentially introduces a priori assumption into the diffusion training process: different images should correspond to different points in the Gaussian distribution. This prior reduces the difficulty of diffusion training, but also brings concerns about practicality and generalization:
    - Practicality: The datasets for unconditioned generation used in this paper are all based on image classification (CIFAR/CelebA/ImagenNet), so these data distributions themselves already meet the above prior assumptions (each class is a kind of image distribution). However, for image datasets with more complex distributions, such as LAION, whether the proposed assignment strategy can still be used is an uncertain question, and this paper also lacks relevant experiments. You can demonstrate that this is not a problem by training unconditional generation experiments on a subset of LAION.
    - Generalization: Since images with the same data distribution are assigned to a specific noise interval during training, this may cause the model to lose the ability that denoising each point in the noise space into any source image during inference, thereby damaging the generalization of the generated images. However, the paper lacks experiments to evaluate the generalization of generated images.

2. Considering that all the unconditional generation results in the paper are trained on classification datasets, is the distribution of noise assigned by the proposed method strongly correlated with the category? Intuitively, after the proposed noise assignment, images of the same category should have closer noise in current batch.

**Questions:**

Please refer to the Weaknesses

---

> ### Author Rebuttal · Authors · 2024-08-07
>
> We are really excited to hear that you agree with our strengths listed above, and we sincerely hope that the proposed perspective of image-noise matching can be realized and further discovered by our diffusion society. We hope to address your concerns below:
>
> **W1-Practicality**
>
> We thank you for the concern on whether immiscible diffusion would work on datasets without simple classification distributions, and we highly appreciate your suggestion on training on LAION. We train conditional Stable Diffusion model on a subset of LAION (laion-10k) with a batch size of 512. We see that after 25k steps, the FID for immiscible and vanilla diffusion is 124.55 and 146.69 respectively, showing that immiscible diffusion can work on the training efficiency of datasets with complex distributions. Note that the training steps are limited by the time and computational resources.
>
> **W1-Generalization**
>
> We can’t agree more to explore on this concern, which we have been seriously considering after submission. A typical example and an important application here is conditional image generation, where noise is randomly sampled and paired with the text prompt. We performed class-conditional image generation on Stable Diffusion with ImageNet dataset. Our results in Fig. R2 showed that immiscible diffusion outperformed the vanilla one, specifically the FID for immiscible and vanilla diffusion is 20.90 and 22.44 respectively, and the training steps necessary to go below the FID <23 is 12.5k for vanilla and 7.5k for immiscible diffusion. This effectively suggests that the generalization problem is not significant for current immiscible diffusion methods, which is not surprising as the distance between image and noise shrinks only ~2% after the assignment. We hope this experiment can help to improve our discussion on generalization.
>
> **W2 Image Correlation according to Classes**
>
> We appreciate your interesting questions, but the answer is no. The images are more correlated according to its own structure rather than the category. In our response to Reviewer y1v3 - W3 - (1), we perform an experiment showing that when enough perturbation is added onto a noise, the image would change to another image in another class, which proves our claim, as shown in Fig. R7. Furthermore, we had an experiment in W1-Practicality on LAION-10k, which nearly has no classifications and immiscible diffusion preliminarily works. This also proves that classification is not a prerequisite for immiscible diffusion.

---

> ### Comment · Reviewer_h6mk · 2024-08-10
>
> Thanks for your rebuttal, here is my reply to your rebuttal:
>
> **W1-Practicality**
>
> Your reply partially addresses my concerns about the effectiveness of this method on complex datasets (rather than simple classification distributions). However, the training data for this experiment is only 10k, and I think the data size should be at least about 10% ImageNet (the dataset used for conditional generation in the paper) to prove its effectiveness.
>
> In addition, could you provide the results of how much the improvement in FID on ImageNet is compared to the improvement on FID on LAION under the same training setting (10k data + 25k training steps)? It would be interesting to explore how much performance differences your method has on training sets with different data distributions.
>
> **W1-Generalization**
>
> The authors may have misunderstood my concerns. FID is good, but it is not an accurate measure of the quality and generalization of generated images. Lower FID does not mean your generative model is better in performance and generalization
>
> * For performance, you can also report CLIP-Score [1] and CMMD [2] for conditional generation (class name as the text prompt).
> * Regarding generalization, I am worried that after training with your method, the model will sample pictures with the same pattern for different random initialization noises, thereby damaging the diversity of generated images. To prove this, you can refer to this representation learning method [3] to see whether the features corresponding to the generated images of the same category are evenly distributed on the feature plane, and calculate their quantized values ​​to determine whether there is enough diversity between the generated images.
>
> [1] Learning Transferable Visual Models From Natural Language Supervision, ICML 2021
>
> [2] Rethinking FID: Towards a Better Evaluation Metric for Image Generation, CVPR 2024
>
> [3] Understanding Contrastive Representation Learning through Alignment and Uniformity on the Hypersphere, ICML 2020
>
> **W2-Image Correlation according to Classes**
>
> I am satisfied with the answers and have no further questions or doubts.
>
> **About Rating**
>
> I tend to keep my score as is so far and will adjust the final score based on further explanations from the authors.

---

> ### Author Response · Authors · 2024-08-13
> **Response to Reviewer h6mk's Comments on Our Rebuttal**
>
> We are sincerely grateful for the reviewer's time and efforts on reviewing our work! We hope to provide more information to clarify our findings on the reviewer's concerns:
>
> **W1 - Practicality**
>
> 1. Larger Complex Image Dataset
>
> We thank the reviewer for raising the concern. We are actively seeking to address this concern via training on large-scale LAION dataset. However, we found that the full LAION dataset has been removed due to the legal policy so we can not download it yet. We can only take the LAION-10K dataset which is legally available on Huggingface "wtcherr/LAION10K". To try our best to do reviewer h6mk a favor, we conduct experiments on the LAION-10K dataset, the results on LAION 10K show that we improve the baseline by 22.14 FID, demonstrating the effectiveness of our method. Despite the relatively small scale, we believe it still effectively shows the sign of life for our method on complex conditional generation.
>
> 2. Comparison Experiments on LAION-10k and on 10k Images in ImageNet
>
> We thank the reviewer for proposing the experiment. We perform an additional experiment on 10k ImageNet data + 25k training steps with vanilla and immiscible conditional SD. The results are nearly the same as the experiments on LAION-10k dataset: the FIDs for vanilla SD on ImageNet (10k images) and LAION are 148.90 and 146.69 respectively, and those for immiscible SD are 128.66 and 124.55 respectively. The FID improvements are 20.24 and 22.14 respectively. The improvement is even a little bit larger on LAION-10k than on ImageNet (10k images), supporting that immiscible diffusion does not lose practicality in complex datasets.
>
> **W1 - Generalization**
>
> We understand the concern on the FID, and hope to present some additional evaluations to address your concerns:
>
> For Performance:
>
> We appreciate the suggestions of providing CLIP-Score and CMMD as other evaluation metrics for evaluating the image-prompt correlation and the image quality, respectively. We evaluate these metrics on 50k images generated from conditional Stable Diffusion models trained for 20k steps, which corresponds to the settings reported in Fig. R2.
>
> The CLIP-Score for baseline and the immiscible model is both 28.55, with standard deviation of 0.02 and 0.01 respectively. We measured it for 3 times as the scores are so close, and our results further validate that there are no differences concerning the CLIP-Score, indicating that the image-prompt correlation is not damaged by immiscible diffusion.
>
> For CMMD, the value for baseline and immiscible model is 1.436 and 1.385 respectively (the lower the better). This also demonstrates that our immiscible diffusion outperforms the vanilla class-conditional SD.
>
> For generalization:
>
> We sincerely thank the reviewer for proposing the additional experiment to prove the diversity of images under the same class-condition. We carefully read the paper provided, and we actively try to design experiments to show the image diversity with representation learning methods. However, the previous representation learning is studied on top of the feedforward models, and we find it unclear how to investigate the visual representation for a multi-step diffusion models. Nevertheless, we have submitted new proofs in Fig. R10, which contains images generated with the same prompt on our immiscible class-conditional SD without cherry picking, to the ACs with anonymous links. It clearly shows that the images are diverse in each class. We hope the release of those additional images can address the reviewer’s concerns.
>
> **W2 - Image Correlation according to Classes**
>
> We thank the reviewer for acknowledging our response, and are sincerely grateful for the reviewer’s efforts in reviewing our work.

---

> ### Comment · Reviewer_h6mk · 2024-08-13
>
> Thanks to the author for his efforts and reply. After reading the author's rebuttal carefully, I decided to increase my score to Weak Accept, but I hope the author can explain in detail the impact of the change on generalization in the subsequent version.
>
> Specifically, you can generate 100 images for each class, then test the image features corresponding to these 100 images, and measure the distance between these image features to determine whether these images are diverse. If the generated images are diverse, it means that there are differences between these image features, and the distance between them is not very close, and there will be no mode collapse phenomenon similar to that in GAN (the features of all images are very close and collapse to a point in the feature space), and I believe t-SNE or the mentioned paper can better help you verify this problem.

---

> > ### Author Response · Authors · 2024-08-13
> > **Response to Reviewer h6mk's Comments on Our Rebuttal - 2**
> >
> > We sincerely appreciate the reviewer's acknowledgement on our response, and we are grateful for all the time and efforts provided by the reviewer. We will implement the reviewer's interesting suggestion in our final version.

---

### Author Rebuttal · Authors · 2024-08-07

Dear Reviewers and ACs,

We are grateful for your time and effort in reviewing our work. We are glad to see that our work is recognized as reasonable (R-h6mk), novel and interesting (R-nEKA, R-y1v3), simple and effective (R-h6mk, R-nEKA, R-y1v3). We are also encouraged to hear that our experiments are acknowledged to show robustness, effectiveness and efficiency (R-y1v3) and our writing is clean and concise (R-h6mk, R-y1v3). We sincerely appreciate the insightful comments and we address the concerns as below:

**Conditional Generation** We perform a class-conditional image generation with Stable Diffusion (SD) on ImageNet-1k with a batch size of 2048. To cater to text-to-image generation, we use class name as the text prompt for SD. Results in Fig. R2 shows that the FID for immiscible class-conditional SD is 20.90, which is 1.53 lower than that of SD baseline. Qualitative comparisons further prove such enhancements, which augment the effectiveness of immiscible diffusions into commonly-used conditional image generation.

**Comparison to OT-CFM and related works** Thanks for pointing it out and we indeed miss this discussion in our submission. We will include them in our final version. Our motivation is different from OT-CFM [1], Multisample flow matching [2] and Approximated-OT [3],  and we target diffusion-based methods while most of these works are for flow-matching based methods. We point out that it is a coincidence that we share a similar algorithmic spirit but we can achieve our motivation -- immiscibility from an orthogonal way. The discussion is presented below:

Theoretical Difference

OT-CFM is designed to “yielding straighter flows that can be integrated accurately in fewer neural network evaluations”[1], while immiscible diffusion aims to keep images immiscible in the noise space. OT focuses more on the straightness of the diffusion path, while immiscible diffusion focuses on the immiscibility of diffusion destinations.
Therefore, minimizing image-noise distance in a mini-batch is only one way to achieve immiscibility. As long as the image’s miscibility is limited, it falls in our motivation. Of course, image-noise OT is one way without adding crossings in the diffusion path.

Additional Experiments Splitting Two Proposals.

To validate the effectiveness of immiscible diffusion alone, we design a controlled variable experiment which does not qualify OT. We still use assignment, but compared to linear assignment calculating distances between images and noises, we flip the dimension of the noise for performing the calculation and assignment. For example, say that the image and the noise are 3072D:  OT-qualified assignment would calculate distance between corresponding dimensions, i.e.

Dist. = sigma(||x0_i - noise_i||2)

Our experiment changes this distance goal into:

Dist.’ = sigma(||x0_i - noise_(3072-i)||2)

By doing this, we still limit the miscibility. However, this does not qualify OT because the real distance is not optimized. In fact, the real distance reduced only <1% in this way.
Interestingly, when we perform the vanilla OT assignment and flip assignment, on DDIM with 50 sampling steps, batch size of 256, no additional image normalization, they show similar performances after initial steps, as shown in Fig. R3, with OT one performs slightly better. This proves that the immiscible effect itself can help improve the diffusion performance. For their performance difference, note that the flip assignment causes crossings in diffusion paths. As shown in Fig. R4, if we have 4 images x0 to x3 and 4 noises n0 to n3, when we flip x and y axis, a path crossing shown in red circle would happen, which adversely affects the FID.

[1] Tong et al. "Improving and generalizing flow-based generative models with minibatch optimal transport."

[2] Pooladian et al. "Multisample flow matching: Straightening flows with minibatch couplings."

[3] Kim et al. “Improving Diffusion-Based Generative Models via Approximated Optimal Transport”

Domination of Immiscible Effect to OT in Noise Assignment

We further hope to understand either immiscible effect or the OT dominates the performance enhancements in image-noise assignment. So we do some stats: As shown in Tab. 3, the distance reduction after assignment is only 2%. We further calculated the stddev of distances in a batch, which is ~10%. Therefore, the distance reduction is so little that, as suggested by R-y1v3, is hard to believe to become the reason for performance enhancement. On the other hand, we calculated the relation between the distance between images and their corresponding noise points, shown in Fig. R5. We see that after assignment, those images far away are assigned to noise points far away, which constitutes the concept of immiscibility. Our results in Fig. 3 also  show that the vanilla diffusion model suffers from the miscible problem while immiscible diffusion significantly reduces it. In conclusion, we believe immiscible diffusion dominates the performance enhancement in image-noise assignment.

Immiscible Diffusion with Image Projection

We notice that simple image projection, i.e. multiple each image by a factor like 2.0 or 4.0, can also perfectly achieve this goal as long as the factor is not too large to wipe out the image in diffusion.

During training, We multiply the images in DDIM, whose original stddev is 0.5, by factors of 2 and 4 to let its stddev be 1.0 and 2.0. Note that all images are centralized so there is no impact on their means. In Fig. R6, we note that after multiplications, the FID significantly improves, which suggests the effectiveness of immiscible diffusion from another angle. Adding assignment can generally further increase the training speed, but the effect would be weakened when the factor is large, as the immiscible problem is significantly solved by the factoring. This offers another way of immiscible diffusion, which will be added in the final version.

---

> ### Comment · Reviewer_Da24 · 2024-08-09
> **Answer to authors’ global rebuttal**
>
> Thank you for your rebuttal. Here is my answer to your global rebuttal:
>
> **Conditional Generation**
>
> The authors experimented on conditional generation by training a Stable Diffusion model to generate ImageNet images conditionally on their class. The authors show the FID score decreases faster with their method, which is encouraging.
>
> However, I have concerns related to this experiment:
> - The authors only show 5 images (potentially cherry-picked) as qualitative examples. This concern was already raised in the initial reviews (Related to reviewer Da24 W4.1 Cherry-picking)
> - It is possible that the model ignores the class name. The results shown in Figure R2 do not show that the generated images have the correct class. Furthermore, consider this thought experiment where you would have only two classes, “bright image” and “dark image”. With the proposed data-noise assignment pair, bright images will only be diffused to the brighter noise samples and dark images will only be diffused to the darker noise samples. Then the model will likely lose its ability to transform a “dark initial noise” (randomly sampled at inference time) into a “bright image” when conditioned on the class “dark image”.  The results shown in the rebuttal do not address this concern. This concern was already raised in the initial reviews (reviewer h6mk W1-Generalization)
> - The main argument mentioned by the authors to support their method is that the first denoising direction predicted by the model (at timestep T) points to the average of the data distribution, which the authors say “does not provide any meaningful information” (L161). By “immiscibility” approach, the first denoising direction now points to the “average of nearby data-points” (L181). While this argument may hold for unconditional generation, it does not make sense anymore as (without immiscibly) it is a good thing to first predict a direction point to the average of images of the class.  This average of images of the class is informative (see https://web.mit.edu/torralba/www/ne3302.pdf Figure 1, see also first column of Figure 5 of https://arxiv.org/pdf/2305.08891 ). Predicting this average in the first denoising step allows to guide the generation towards images of the desired class.
>
> **Comparison to OT-CFM and related works**
>
> The argument given by authors in the rebuttal are not convincing:
>
> - To differentiate from OT-CFM, the authors now propose to redefine immiscibility as “immiscibility of diffusion destinations without necessarily minimizing image-noise distance in a mini-batch”, in other words: coupling (CFM) but no optimal transport (OT-CFM). However, this new definition still makes the proposed method very similar to CFM. The explanation in **Theoretical Difference** only explains the difference between OT-CFM and normal CFM. This concern was already mentioned in the initial reviews (Da24, W2.1 Coupling).
> - **Additional Experiments Splitting Two Proposals/Domination of Immiscible Effect to OT in Noise Assignment**. The paper [1] (referenced in the rebuttal) already showed that CFM (new definition of “immiscibility”) can have better FID that standard FM.
> - **Immiscible Diffusion with Image Projection**: Multiplying all pixels by a factor of 2 or 4 in the training preprocessing is equivalent to reducing all noise levels by this factor. Some studies (https://arxiv.org/abs/2305.08891, https://www.crosslabs.org/blog/diffusion-with-offset-noise, https://arxiv.org/abs/2309.15842 ) showed however that the noise level used in Stable Diffusion (v1 and v2) are already too low, preventing the model to generate non-greyish images. The paper https://arxiv.org/pdf/2305.08891 showed (table 3) that increasing the noise levels fixes this issue and improve the FID score. I am very surprised that the authors here (Fig R6) find that decreasing all noise levels by a factor of 2 or 4 also improves the FID. Some discussion would be needed here, at least by showing qualitative examples.

---

> > ### Author Response · Authors · 2024-08-11
> > **Rebuttal to Reviewer Da24's comment on global responses**
> >
> > We thank reviewer Da24 for sharing valuable time with us and great efforts for reviewing and commenting on our rebuttals! Our further comments are attached below:
> >
> > 1. Conditional generation:
> >
> > 1) We emphasize that our quantitative results have already shown that immiscible diffusion improves 1.53 FID of conditional generation with SD (https://openreview.net/forum?id=kK23oMGe9g&noteId=8Wj7beyL9K). The qualitative results in Fig R2 are cherry-picked but given the quantitative improvement over vallina baseline, the visualization is used for supporting quantitative results and helps the audience understand better how our method improves the baseline method. Moreover, we refer the reviewer Da24 to check Fig 7 in our submission, the unconditional generation results on ImageNet are not cherry-picked. While there might be a few cases that do not show significant visual improvement, many examples like the 1st row 4th column, 2nd row 5th column, 3rd row 5th column etc. are apparently better than the baseline. Due to the fact that we have only 1 page for images, we were not able to put non-cherry-picked results during the rebuttal. We have now packed tens of classes of non cherry-picking images in an anonymous link that has been sent to the ACs for approval to show.
> >
> > 2) Thanks for the question from the reviewer Da24. We do not observe this phenomenon in our experiments. Our method does not hurt the conditional generation and results in the conjectures from the reviewer Da24. We randomly checked 4 classes, each with 5 images without cherry-picking, seeing that 1) these images correspond to their prompt class; 2) in each class, the images enjoy a sufficient diversity. We have concluded these results as Fig. R10, which has been sent to the ACs for approval to show. In the anonymous link we also include all ~50 images in each class checked for the reviewers’ information.
> >
> > 3) Thank the reviewer Da24 for the comment. In our experiments, immiscible conditional SD is indeed capable of performing better than vanilla one. Therefore, we still believe that immiscible diffusion works better in conditional generation cases. While the reviewer Da24 claims that “it is a good thing to first predict a direction point to the average of images of the class”, we kindly cannot find it excludes the possibility that our method is better.
> >
> > 2. Comparison to OT-CFM
> >
> > We sincerely appreciate the reviewer Da24 brings up flow-matching based methods during discussion. We emphasize that immiscible diffusion is a method upon diffusion models, and OT-CFM is built upon the framework of flow-matching method. Beyond it, we respond to the concern from Da24 as below.
> >
> > 1) We respectfully disagree that we are similar to coupling in I-CFM. To our knowledge, the random coupling in I-CFM in [1] does not qualify as an immiscible diffusion; also, our additional flip assignment design cannot be found in paper [1]. OT-CFM [1] makes the image-noise closer but our immiscible diffusion highlights they should be separated to each other, the flip assignment experiments demonstrate the effectiveness and the differences of our method. We believe this high-level motivation is different and novel.
> >
> > 2) We politely point out that we don’t see the effect claimed by the reviewer in paper [1]: We see that in the paper [1] Fig. 3 left, it clearly shows that only with OT the training efficiency be improved, the OT contributes to the majority of FID improvement. Furthermore, this is a work based on FM, which seems to be out of our scope in diffusion methods.
> >
> > 3) We very kindly suggest the reviewer noting that we are using DDIM, not SD, in this experiment.

---

> > > ### Comment · Reviewer_Da24 · 2024-08-14
> > >
> > > **1.1.**
> > >
> > > I do not deny that the FID plot is improved in the reported results for this experiment, but FID alone might not be sufficient for evaluation.
> > >
> > > I do not agree that cherry picking “helps the audience understand better how our method improves the baseline method”, as it shows improvements better than one can expect in average.
> > >
> > > Figures 7 to 11: I do not agree with “there might be a few cases that do not show significant visual improvement”. From what I can see, most cases do not show **significant** visual differences.
> > >
> > > **1.2. / 1.3.**
> > >
> > > This would need to be explored in more details. I find it a bit weird that it is possible to claim that noising is immiscible (images are only noised locally) but that the model is able to do miscible denoising (model is still able to denoise any noise sample to any class). I suggest to look at the simple case I suggested with bright / dark images, and expand these explanations
> > >
> > > **2.1. / 2.2**
> > >
> > > I-CFM uses independent coupling, meaning the samples from prior (noise x_1) and target (data x_0) distributions are sampled independently, which does not qualify as “immiscible”.
> > > It seems that the CFM formulation allows to select the coupling by a joint distribution q(z) = q(x_0, x_1) which can be arbitrary (not necessarily independent or OT: for instance SB-CFM, EB-CFM). The CFM paper (Arxiv 2023.02 / ICML 2023 workshop / TMLR 2024) also showed better metrics for non-OT and non-independent couplings (training time Table D1, Wasserstein distance Figure D6, although, yes, no FID are reported for that case)
> > >
> > > **2.1 / 2.2**
> > >
> > > I disagree that “high-level motivation is different and novel”. Furthermore, it seems that the CFM paper also includes “objective variance” analysis (D.1), which seems to be a way to quantify immiscibility.
> > >
> > > **2.3**
> > >
> > > Thanks for the clarification. The update rule in the diffusion reverse process is designed such that the distributions of x_t in the reverse process and in the forward process match for every t. This implicitly assumes x_T is indistinguishable from noise during training. Decreasing the maximal noise level can hurt this assumption.

---

### Decision · Program_Chairs · 2024-09-25

**Decision:**

Accept (poster)

**Comment:**

The paper introduces "Immiscible Diffusion," a method to accelerate diffusion model training by refining noise-data mapping. Inspired by the concept of immiscibility in physics, the method assigns specific noise to images to simplify the optimization process, leading to faster training with maintained image quality. The reviewers agree that this approach requires only a minor code change to achieve significant speed improvements, and it speeds up training while maintaining high-quality image generation. On the other hand, the reviewers also raised the concerns that the paper still does not provide very strong comparisons with existing methods and the experimental validation is not robust enough. After carefully reviewing the paper, reviewer discussions, and rebuttals, the AC recommends accepting the paper.